# Maternal dietary fat during lactation shapes single nucleus transcriptomic profile of postnatal offspring hypothalamus in a sexually dimorphic manner in mice

Yi Huang [1,8,9], Anyongqi Wang [1,2,9], Wenjiang Zhou[3], Baoguo Li[1,4], Linshan Zhang[3], Agata M. Rudolf[1], Zengguang Jin[5], Catherine Hambly [6], Guanlin Wang [3,9] ✉ & John R. Speakman [1,5,6,7] ✉

Maternal overnutrition during lactation predisposes offspring to develop metabolic diseases and exacerbates the relevant syndromes in males more than females in later life. The hypothalamus is a heterogenous brain region that regulates energy balance. Here we combined metabolic trait quantification of mother and offspring mice under low and high fat diet (HFD) feeding during lactation, with single nucleus transcriptomic profiling of their offspring hypothalamus at peak lacation to understand the cellular and molecular alterations in response to maternal dietary pertubation. We found significant expansion in neuronal subpopulations including histaminergic (Hdc), arginine vasopressin/retinoic acid receptor-related orphan receptor β (Avp/Rorb) and agouti-related peptide/neuropeptide Y (AgRP/Npy) in male offspring when their mothers were fed HFD, and increased Npy-astrocyte interactions in offspring responding to maternal overnutrition. Our study provides a comprehensive offspring hypothalamus map at the peak lactation and reveals how the cellular subpopulations respond to maternal dietary fat in a sex-specific manner during development.

Obesity has increased rapidly in the past five decades with currently more than 2.5 billion people being overweight or obese[1]. Emerging evidence suggests that maternal nutrition during pregnancy or lactation is crucial for offspring growth and development[2–8]. The impact of maternal diet exposure may interact with overnutrition in later life to exacerbate offspring adiposity. Previous studies[9,10] demonstrated that

lactation is an important period for maternal effects since during lactation offspring rely solely on the mother's milk for their nutrition and growth. In both human and animals, maternal overnutrition during pregnancy and/or lactation predisposes offspring to obesity, and increases the risk of metabolic syndrome, insulin resistance and hypertension in offspring[6,8,11–18]. However, these effects may vary

[1]State Key Laboratory of Molecular Developmental Biology, Institute of Genetics and Developmental Biology, Chinese Academy of Sciences, Beijing 100101, China. [2]University of Chinese Academy of Sciences, Beijing 101408, China. [3]Shanghai Key Laboratory of Metabolic Remodeling and Health, Institute of Metabolism and Integrative Biology, Centre for Evolutionary Biology, Fudan University, Shanghai 200438, China. [4]Tianjian Laboratory of Advanced Biomedical Sciences, Zhengzhou University, Zhengzhou 450001, China. [5]Shenzhen Key Laboratory of Metabolic Health, Center for Energy Metabolism and Reproduction, Shenzhen Institutes of Advanced Technology, Chinese Academy of Sciences, Shenzhen 518055, China. [6]School of Biological Sciences, University of Aberdeen, Aberdeen AB24 3FX, UK. [7]China Medical University, Shenyang, Liaoning 110122, China. [8]Present address: Broad Institute of MIT and Harvard, Metabolism Program, Cambridge, MA 02142, USA. [9]These authors contributed equally: Yi Huang, Anyongqi Wang, Guanlin Wang. ✉e-mail: guanlin_wang@fudan.edu.cn; j.speakman@abdn.ac.uk

depending on the sex of the offspring with male offspring being more sensitive[8,14,18–22]. Although numerous studies have studied such effects in both sexes[23–25], the majority of previous studies primarily focused on male offspring[12–14,16]. Since at least half of the world population is female, understanding similarities and differences in maternal diet effects on both sexes is an important goal.

The hypothalamus regulates energy balance and other vital physiological processes, including hormone release from the pituitary, temperature regulation, circadian rhythms and social/sexual behaviors[26–29]. Neurons, glial and immune cells with diverse cellular functions organize into spatially distinct but structurally contiguous nuclei[30–34]. In the hypothalamic arcuate nucleus (ARC), canonical neuropeptides agouti-related peptide (AgRP) and pro-opiomelanocortin (POMC) play central roles in regulating food intake and energy homeostasis[35–39]. Offspring raised by mothers with maternal obesity or exposure to a high fat diet (HFD) during pregnancy and/or lactation usually have a higher ratio of AgRP/POMC expressing neurons[5,8,40–42]. Previous studies of the transcriptomic profiles of the offspring hypothalamus in response to maternal overnutrition were addressed at the tissue level using bulk RNA-seq[43–45], which makes it difficult to reveal the complicated hypothalamic heterogeneity, key functional pathways and cellular communications between cell subpopulations. Single cell/nucleus transcriptomics is a powerful technique that enables the dissection of the cellular heterogeneity and pathways in the offspring hypothalamus in response to maternal overnutrition. Multiple studies have recently employed single cell/nucleus RNA-seq to better understand the development of the rodent hypothalamus[44,46–58]. The 'HypoMap' integrated multiple datasets and generated the largest single cell gene expression atlas of the murine hypothalamus to date[59]. However, no previous study profiled the postnatal offspring transcriptomic landscapes in the hypothalamus to understand the impact of maternal nutrition during lactation at the single cell level, which may enhance our understanding of the sex-specific offspring response to the maternal diet. This could facilitate the development of interventions to reduce the risk of obesity in offspring when exposed to an unhealthy maternal diet.

Here we performed maternal and offspring physiological measurements of mice exposed to maternal high fat diet (HFD) or low fat diet (LFD) during lactation and generated a single nucleus transcriptomic (snRNA-seq) atlas of 38,594 high quality nuclei after quality control of the male and female offspring hypothalamus during peak lactation (age postnatal day 15, P15). Mothers fed with HFD had significantly elevated milk energy output than those fed with LFD. Their offspring were also significantly heavier than those from the maternal LFD group. We identified nine key cell populations including neurons, astrocytes, tanycytes, ependymal cells, oligodendrocyte precursor cells and oligodendrocytes, microglia, stromal cells, and immune cells including macrophages and B cells in the offspring hypothalamus. Histaminergic (Hdc) and arginine vasopressin/retinoic acid receptor-related orphan receptor β (Avp/Rorb) neuronal subpopulations and the well-known AgRP neurons were enriched in the male offspring specifically when their mothers were fed HFD, highlighting that maternal dietary exposure during lactation has significant effects on neurogenesis, particularly in male offspring, which may contribute to their later susceptibility to obesity when exposed to HFD themselves. Although previous work has highlighted differential expansion of AgRP neuron populations in males under maternal HFD feeding[5,8,40–42], expansion of Avp/Rorb and Hdc cell populations has not been previously suggested. We found increased cellular interactions in male offspring when their mothers were fed HFD, compared to the other groups, suggesting extensive modulation resulting from overnutrition in male offspring. We further experimentally validated that there were increased neuronpeptide Y (Npy) neuron-astrocyte interactions in the offspring responding to maternal HFD. Our findings provide a comprehensive atlas of the mouse offspring hypothalamus

at P15 (peak lactation of their mothers) responding to mothers fed with different levels of dietary fat[60], and unravel how the cellular subpopulations respond to overnutrition in a sex-specific manner. We have made the dataset publicly available (https://mouse10x.shinyapps.io/p15atlas/).

## Results

In this study, we integrated physiological measurements from both mothers and their offspring to investigate the impact of maternal diet during lactation on energy balance and utilized single nucleus transcriptomic sequencing to gain insights into the alterations occurring in the offspring hypothalamus driven by maternal diet.

### Higher energy intake in mothers fed HFD during lactation

12-Week old female C57BL/6 N mice were fed with LFD for a 2 week baseline period and during pregnancy. After birth (day 0), mothers were fed LFD ($n = 9$) and HFD ($n = 6$) across the lactation period (diet swapped from lactation day 1) (Fig. 1a). There were no significant differences observed in maternal body weight (BW) between LFD and HFD groups before mating (baseline) (ANOVA, $F = 0.021$, $P = 0.887$), during late pregnancy (7 days before parturition) (RM GLM, $F = 0.133$, $P = 0.722$) and during lactation (RM GLM, $F = 0.398$, $P = 0.539$) (Fig. 1b, Supplementary Data 1). A significant effect of day of lactation (RM GLM, $F = 50.992$, $P < 0.001$) on maternal BW was observed during lactation (Fig. 1b), reflecting a significant increase in the maternal BW across the lactation period in both maternal LFD and HFD groups (Fig. 1b). In addition, no significant differences were observed in body fat mass and lean mass before mating (ANOVA, fat mass: $F = 0.636$, $P = 0.439$; lean mass: $F = 0.944$, $P = 0.349$) on lactating day 1 (ANOVA, fat mass: $F = 0.039$, $P = 0.847$; lean mass: $F = 0.213$, $P = 0.652$), day 10 (ANOVA, fat mass: $F = 0.469$, $P = 0.505$; lean mass: $F = 0.549$, $P = 0.472$) and day 16 (ANOVA, fat mass: $F = 0.063$, $P = 0.807$; lean mass: $F = 0.006$, $P = 0.937$) (Fig. 1c, d).

There was no significant difference in maternal gross food intake (FI) before mating (ANOVA, $F = 0.346$, $P = 0.567$) and during late pregnancy (RM GLM, $F < 0.001$, $P = 0.998$) between groups when they were all eating the LFD. Between lactation days 1 to 15 there was a highly significant effect of day of lactation (RM GLM $F = 35.18$, $P < 0.001$) as well as a significant effect of diet ($F = 5.134$, $P = 0.041$) on maternal gross FI. Mothers fed HFD had a lower daily FI (by weight) than those fed LFD (Fig. 1e, Supplementary Data 1). The gross energy content of the foods was 18.234 and 23.101 KJ/g for 10% and 45% fat diets respectively. Significant effects of day of lactation (RM GLM, $F = 37.967$, $P < 0.001$) and diet ($F = 10.192$, $P = 0.007$) were also observed on gross food energy intake ($GE_{food}$) during lactation. Contrasting the observation on the weight of food eaten, mothers fed HFD had a higher daily energy intake than those fed LFD (Fig. 1f, Supplementary Data 1).

There was no significant difference in daily energy expenditure (DEE, measured by doubly-labeled water between lactation days 13 to 15) (GLM, diet: $F = 2.766$, $P = 0.131$; BW: $F = 2.135$, $P = 0.182$) and metabolizable energy intake (MEI) (GLM, diet: $F = 3.175$, $P = 0.108$; BW: $F = 0.722$, $P = 0.42$) between maternal dietary groups after adjustment for BW (Fig. 1g, Supplementary Data 1). The mothers fed HFD had higher milk energy output (MEO) (GLM, diet: $F = 11.941$, $P = 0.007$; BW: $F = 0.145$, $P = 0.713$) than those fed LFD. The MEO from the mice fed HFD (134.23 kJ/d) was approximately 36% more than the LFD group (86.22 kJ/d) (Fig. 1g, Supplementary Data 1).

### Offspring raised by mothers fed HFD during lactation are heavier

There were significant differences between days of lactation ($F = 618.94$, $P < 0.001$) and a significant day×diet interaction ($F = 6.709$, $P = 0.006$) on litter mass (LM) in both maternal dietary groups. There was no maternal dietary effect on the LM before peak lactation (day 1

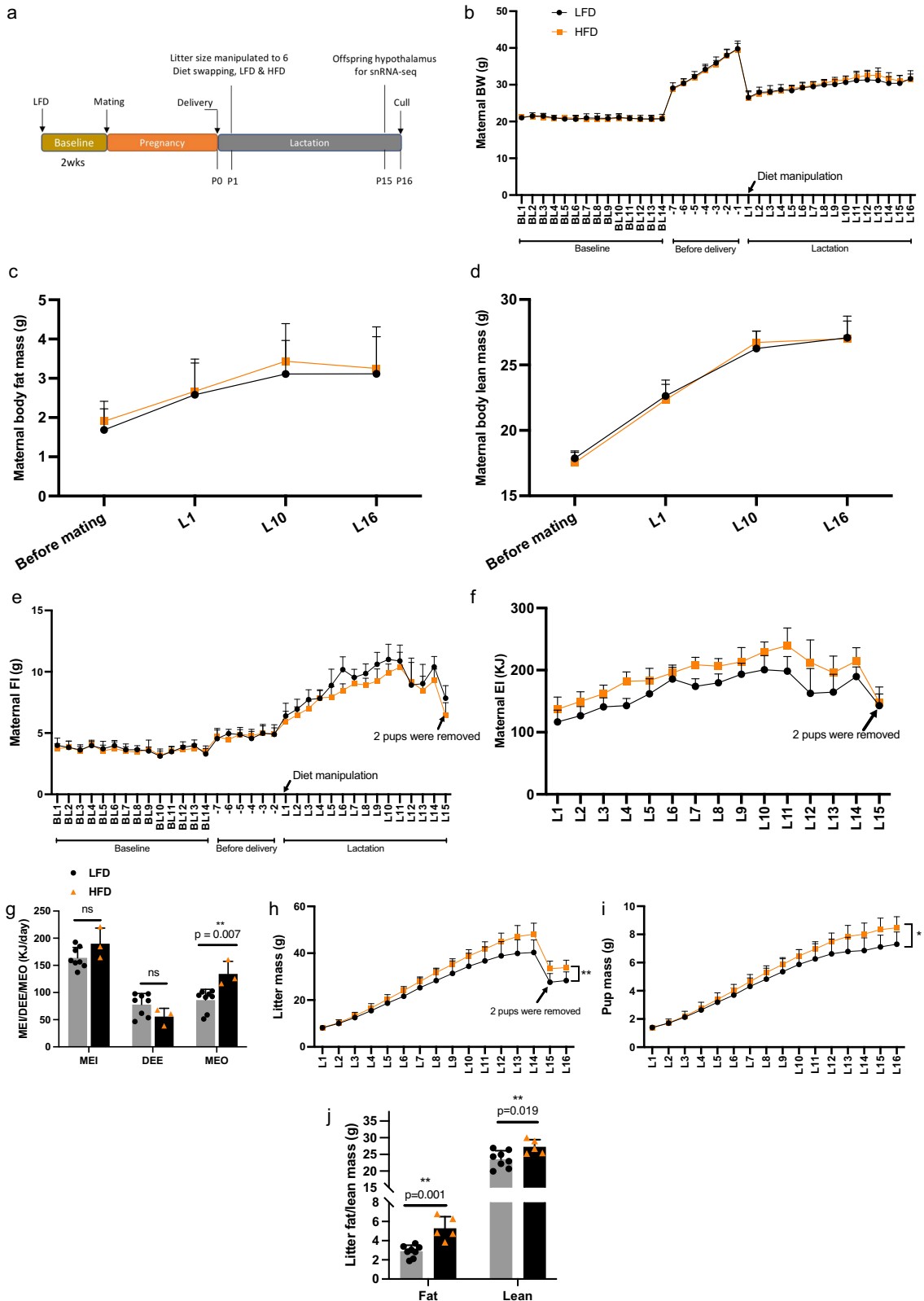

to 11) ($F = 2.044$, $P = 0.176$). During peak lactation, a significant increase in the LM was observed in the litters of mothers fed the HFD (diet: $F = 7.521$, $P = 0.017$). The pups raised by mothers fed HFD had significantly greater LM than those fed LFD (Fig. 1h). Similarly, pup mass (PM) of offspring raised by mothers fed HFD was significantly increased over peak lactation (days 12–16) compared with those fed LFD (RM GLM: diet: $F = 5.89$, $P = 0.031$; day of lactation: $F = 56.449$,

$P < 0.001$; day×diet: $F = 2.636$, $P = 0.093$) (Fig. 1i). Final LM and PM at P16 showed the same patterns of significance in different dietary groups (LM: $F = 9.069$, $P = 0.01$; PM: $F = 7.068$, $P = 0.02$). Offspring from HFD fed mothers were 5.57 g heavier in LM and 1.17 g heavier in PM compared with those raised by mothers fed LFD (Supplementary Data 1). The fat and lean contents of the litters raised by HFD fed mothers were both significantly higher than the LFD ones (ANOVA, fat

**Fig. 1 | Metabolic measurements for mothers fed with different dietary fat during lactation as well as their offspring. a** Diagram of experimental design. **b** Maternal body weight (BW) (**c**) body fat mass, **d** body lean mass, **e** food intake (FI) and (**f**) energy intake (EI) from low fat and high fat dietary groups during lactation. Sample sizes of mothers fed low fat diet (LFD) during lactation were 9, for those fed high fat diet (HFD) were 6. **g** Maternal metabolizable energy intake (MEI), daily energy expenditure (DEE) and milk energy output (MEO) during peak lactation. Sample sizes of maternal LFD group for the measurements were 8, for those from maternal HFD group were 3. *p* value by GLM adjusted with BW. **h** Litter and (**i**) pup mass of offspring raised by mothers fed with different dietary fat during lactation. **j** Litter fat and lean mass of postnatal day16 offspring raised by mothers fed with different dietary fat. Sample sizes of litters from maternal LFD group were 8, for those from maternal HFD group were 5. *p* value by one-way ANOVA. * represents *p* < 0.05, ** represents *p* < 0.01. ns represents no significance between comparisons. Values are means±s.d. LFD represents low fat diet, HFD represents high fat diet, P represents postnatal in (**a**). BL represents baseline, L represents lactation in (**b**–**i**). Source data are provided as a Source Data file.

mass: $F = 22.505$, $P = 0.001$, lean mass: $F = 7.506$, $P = 0.019$). The litters from HFD fed mothers had 45.37% higher fat mass and 13.74% greater lean mass than those fed LFD (Fig. 1j, Supplementary Data 1).

In male offspring, masses of the subcutaneous fat and spleen were 57% and 32% heavier when raised by HFD fed mothers compared to those fed LFD (subcutaneous fat: $F = 18.983$, $P = 0.001$; spleen: $F = 12.877$, $P = 0.004$). A significant maternal dietary effect on subcutaneous fat in male offspring was still observed even after correcting for total BW (subcutaneous: diet, $F = 11.127$, $P = 0.007$; BW, $F = 35.965$, $P < 0.001$) (Supplementary Fig. 1a). Significant maternal dietary effects were also observed in the masses of subcutaneous fat and spleen, as well as in BAT and heart, in female offspring (subcutaneous fat: $F = 10.024$, $P = 0.009$; spleen: $F = 5.732$, $P = 0.036$; BAT: $F = 6.485$, $P = 0.027$; heart: $F = 13.408$, $P = 0.004$). However, after adjusting for total BW, the significance of the differences disappeared except for the mass of the heart (diet, $F = 5.994$, $P = 0.034$; BW, $F = 8.016$, $P = 0.018$), with 20.73% larger heart mass in females from the maternal HFD group (Supplementary Fig. 1b). These results suggested that male offspring were more sensitive in response to maternal diet in particular in fat mass.

## A high-resolution cellular atlas of the offspring hypothalamus at peak of lactation

To enable a comprehensive understanding of the cellular landscape of sex specificity in different cell populations in the postnatal offspring hypothalamus, we generated a single nucleus atlas at peak lactation (offspring P15). We isolated nuclei from female offspring raised by mothers fed LFD (abbreviated as fLFD below) and HFD (fHFD), as well as male offspring raised by mothers fed LFD (mLFD) and HFD (mHFD) (Fig. 2a). A total of 38,594 nuclei were included in the downstream analyses after quality control, doublet removal and batch correction, including 19,159 and 19,435 nuclei from female and male offspring respectively (Supplementary Fig. a, b, Supplementary Data 2). We identified nine major cell types (Fig. 2b–d) annotated by canonical marker genes, cell type specific genes and SingleR[61]. Neurons (marked by *Snhg11*, *Meg3*) comprised 66.3% of the total nuclei followed by non-neuronal cells including astrocytes (9.7%, *Ntsr2*), oligodendrocyte precursor cells and oligodendrocytes (OPCs and ODCs, 8.9%, *Pdgfra*, *Plp1*), stromal cells (7.4%, *Col3a1*), tanycytes and ependymal cells (4.5%, *Col23a1* for tanycytes *and Dnah12* for ependymal cells), endothelial cells (0.4%, *Flt1*), interneurons (0.6%, *Chga*, *Cga*), microglia/macrophages (2.1%, *Siglech*, *Mrc1*) and a tiny population (0.2%) of B cells (*Cd74*, *Cd79a*) (Supplementary Data 3–4).

In addition, we quantified and compared the percentage of each cell type in the hypothalamus across the fHFD, fLFD, mHFD, and mLFD groups as well as individual samples. There were no overall significant differences in terms of the percentage of nuclei between the four groups and cellular composition was similar across the replicates (Fig. 2e, Supplementary Fig. 2c). The neuronal population was consistently the largest in all four groups as in the integrated data (fHFD = 67.8%, fLFD = 68.7%, mHFD = 65.6%, mLFD = 63.3%). We further adapted DAseq[62], a differential abundant cell subpopulations detection method and found that astrocytes and OPCs were expanded in the fHFD group compared with the mHFD group, however, there were no compositional differences between the fLFD and mLFD

groups, nor between the mHFD and mLFD groups(Supplementary Fig. 2d). Differentially expressed gene (DEG) analyses between different groups revealed that the top 3 maternal dietary perturbed cell populations are neurons (5289 DEGs across all the comparisons), astrocytes (398 DEGs) and oligodendrocytes (288 DEGs) (Supplemental Data 4). Top/bottom 5 DEGs ordered by log2 fold change of each comparison in neurons, astrocytes and oligodendrocytes were shown in Supplementary Fig. 2e and/or Supplemental Data. 4. Astrocytes and oligodendrocytes are two important gila cells that regulate neuronal synaptic structure and plasticity in the central nervous system[63]. *Rorb* is functionally related to astrocyte maturation and might be critical for the maintenance of normal neuronal excitability[64]. We observed that *Rorb* hits as one of the top DEGs in maternal HFD feeding groups in astrocytes (mHFD vs fHFD, $P_{adj} = 0.002$; mHFD vs mLFD, $P_{adj} < 0.001$). GO enrichment analyses were performed using the DEGs between different dietary groups in the top 3 perturbed cell populations (Supplemental Data 4). GO-BP pathways such as "myelination" or "regulation of myelination" are significantly enriched in oligodendrocytes between the comparisons of maternal HFD vs LFD or male vs female offspring. The induction of hypothalamic myelin disruption was observed when mice were exposed under chronic HFD feeding[65]. Pathways such as "regulation of neuron death" are significantly enriched in neurons between different maternal dietary groups. HFD induced apoptosis of hypothalamic neurons was observed in rodents[66].

To compare our dataset with the published integrated adult murine hypothalamic dataset HypoMap[59], we projected the cells in our dataset onto the HypoMap dataset using the Symphony[67] reference mapping algorithm and confirmed that the annotation in our dataset is consistent with the HypoMap results in the adult mouse hypothalamus (Supplementary Fig. 2f). This indicated that cell types observed in adult mice are already committed in the newly developed hypothalamus at postnatal day 15. Our dataset therefore represents a comprehensive cellular atlas of the female and male offspring hypothalamus at P15.

## Transcriptional heterogeneity of neurons in the postnatal day 15 offspring hypothalamus

To understand cellular heterogeneity in the neurons, the largest and most diverse cell type in the newly developed hypothalamus, in response to different maternal diets during lactation in both male and female offspring, we sub-clustered the 25,570 neuronal cells into 30 neuronal subpopulations and separated them into excitatory glutamatergic (*Slc32a1*, *Gad1*, *Gad2*), inhibitory gamma-aminobutyic acidergic (GABAergic) (*Slc17a6*) and histaminergic (*Hdc*) neurons (Fig. 3a, b). The neuronal subclusters present 29 subpopulations annotated by known markers and one unassigned subtype. Then we colored the neurons according to the hypothalamic regions including the ARC, the dorsomedial hypothalamic nucleus (DMH), the lateral hypothalamic area (LHA), the paraventricular hypothalamic nucleus (PVH), the supraoptic nucleus (SON) and the ventromedial nucleus of the hypothalamus (VMH) etc (Supplementary Fig. 3a). A list of marker genes was identified for each sub-cluster as shown in Supplementary Data 4. We also examined the gene expressions of some well-known neurotransmitters and neuropeptides in our dataset and we were able

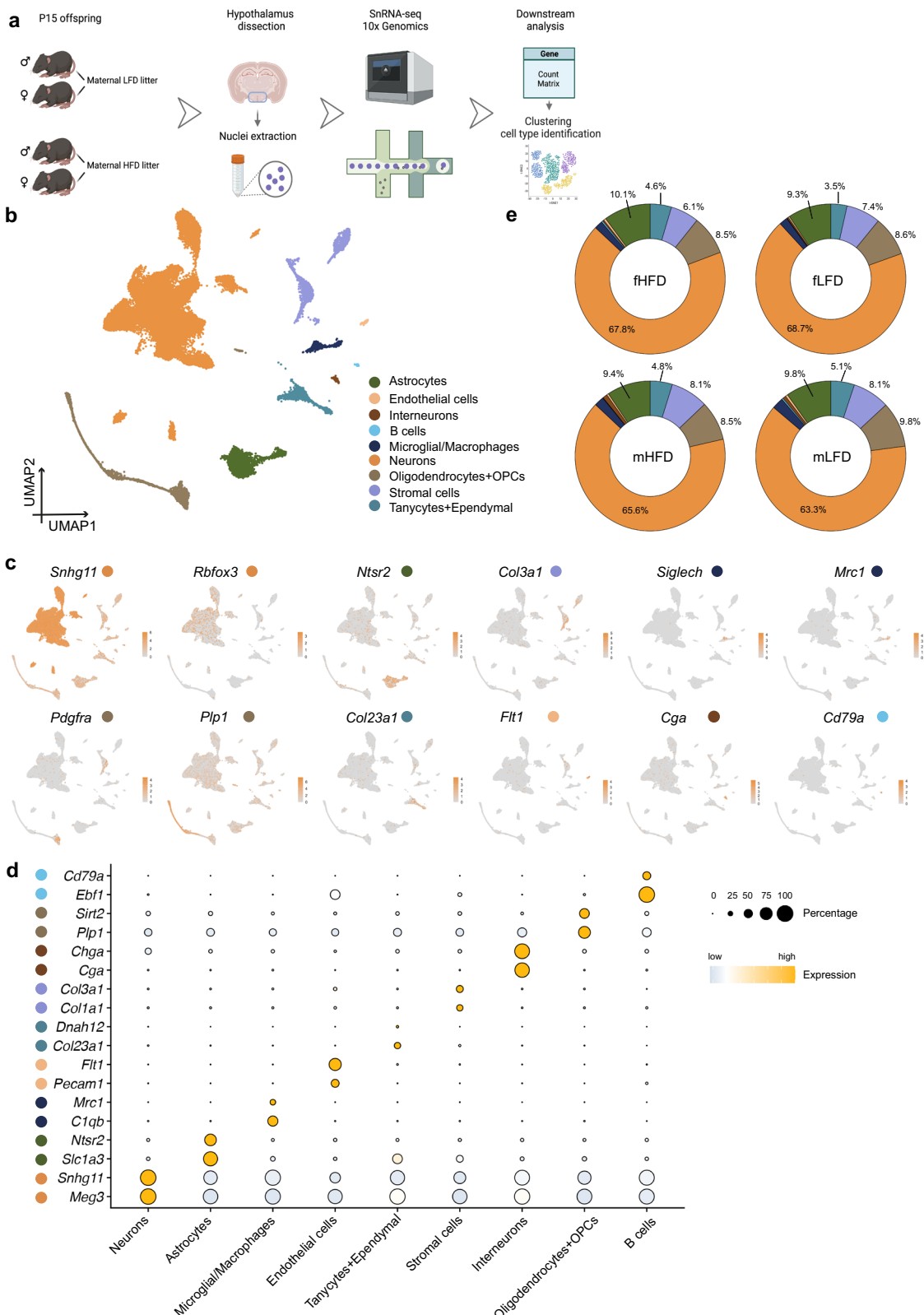

**Fig. 2 | Establishing a single nucleus atlas of mouse hypothalamus from post-natal day 15 (P15) male and female offspring raised by mothers fed with different dietary fat during lactation. a** Diagram of single nucleus RNA-seq experiment using 10x Genomics ($n = 2$ for each maternal dietary group, groups including: fLHD female offspring under maternal low fat diet during lactation, fHFD, female offspring under maternal high fat diet during lactation, mLFD male offspring under maternal low fat diet during lactation, mHFD male offspring under maternal high fat diet during lactation). This image was adapted and created with BioRender.com. **b** Uniform Manifold Approximation and Projection (UMAP) visualization of key lineages after data integration, colored by different cell types (total cell number after QC = 38,594). **c** Canonical marker genes for key lineages shown on the UMAP. **d** Dotplots show the marker genes for key lineages. **e** Donut plots of the cell proportions of key lineages from female and male offspring under maternal LFD and HFD exposure.

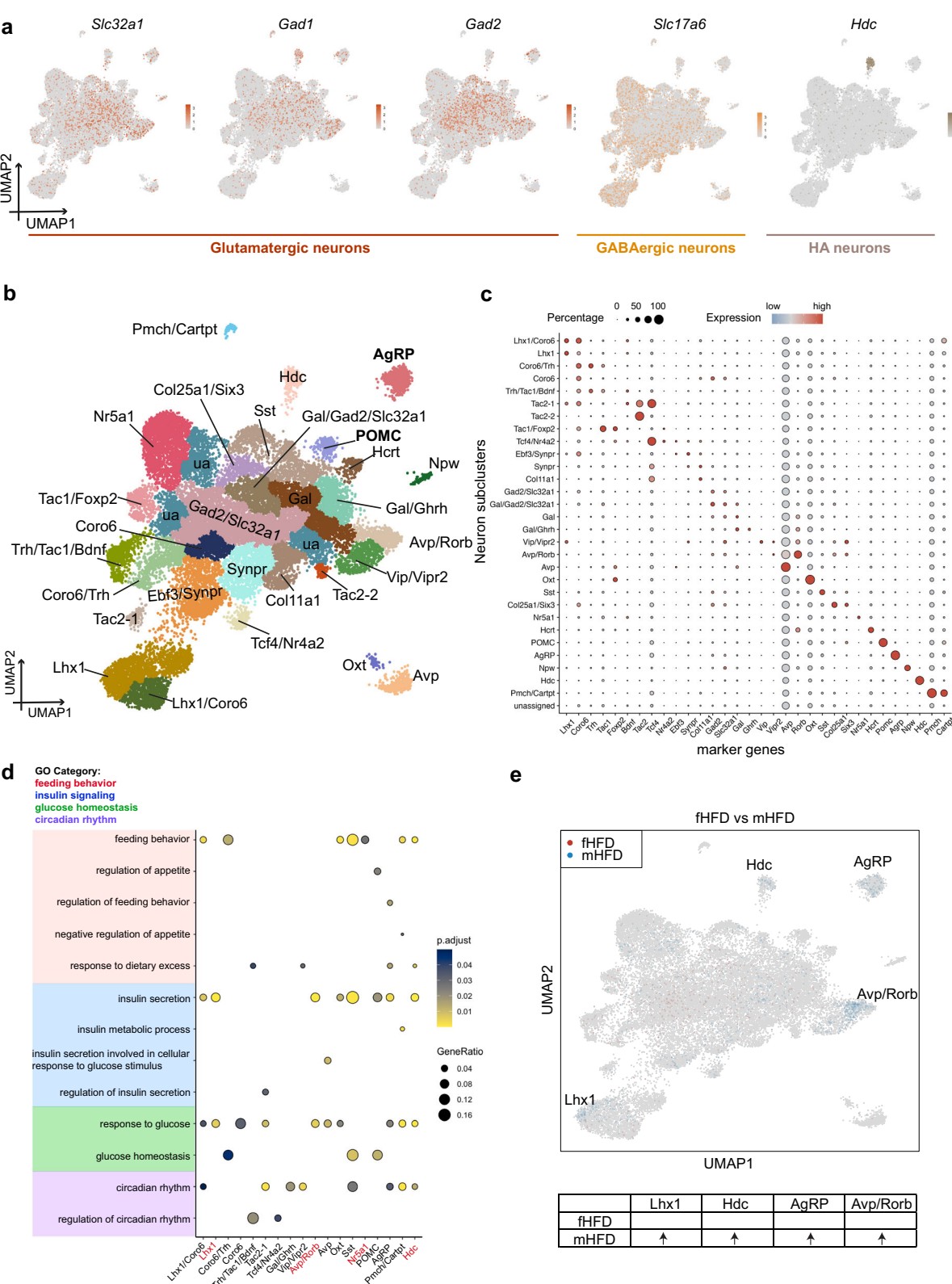

**Fig. 3 | Heterogeneity of P15 offspring hypothalamic neuronal subpopulations.**
**a** UMAP of canonical maker genes of glutamatergic, gamma-aminobutyric acid-ergic (GABAergic) as well as histaminergic (HA) neurons. **b** UMAP of neuronal subpopulations, colored by different neuronal subpopulations. **c** Dotplot shows the canonical marker genes for neuronal subpopulations. **d** GO enrichment analysis of cell type specific transcriptomic signatures shows significant enrichment

$(P_{adj} < 0.05)$ in metabolic relevant categories of feeding behavior, insulin signaling, glucose homeostasis and circadian rhythm in several neuronal subpopulations. $p$ value was adjusted by Benjamin-Hochberg correction. **e** Differential abundance analysis (DAseq) shows the enrichment of the abundance of cells from different neuronal subpopulations when comparing fHFD vs mHFD. Colored dots represent the enrichment of abundance in different groups.

to identify key neuron clusters encoding neuropeptides controlling food intake including Agrp, Pomc, Cocaine and amphetamine-related transcript (Cartpt), cholecystokinin (Cck), neuropeptide Y (Npy), galanin (Gal), hypocretin/orexin (Hcrt) etc (Fig. 3c, Supplementary Fig. 3b). As expected, all the neuron subpopulations were present in all four groups (Supplementary Fig. 3b). To better understand the heterogeneitic metabolic functional roles in different neuronal subpopulations, we performed GO pathway enrichment analysis using cell-type specific marker genes and identified 18 neuronal subpopulations, including AgRP and POMC neurons (the key regulators of food intake and energy homeostasis), were enriched in the metabolic related pathways of "feeding behavior", "insulin signaling", "response to glucose" and "circadian rhythm" categories (Fig. 3d).

We next investigated the differential abundance across the neuronal subpopulations for the four groups using DAseq and validated using scCODA[68], we found an increased number of AgRP neurons in the offspring from maternal HFD groups compared to LFD ones in both sexes (Fig. 3e and Supplementary Fig. 3d). We further found an expansion of the number of AgRP neurons in mHFD group vs fLFD group. Arginine vasopressin (Avp)/retinoic acid receptor-related orphan receptor β (Rorb) neurons were also expanded in the mHFD group compared with the other three groups. The transcriptomic signatures of Avp/Rorb neurons were enriched in the GO-BP "insulin secretion" and "response to glucose" terms (Fig. 3d), in line with the previous finding that AVP neurons modulate diverse metabolic functions including glucose homeostasis[69], insulin secretion[70] and feeding behavior[69]. We also observed increased Hdc neurons in the mHFD group compared with the fHFD group. The enriched pathways of the transcriptomic signatures of Hdc neurons were found to play important roles in feeding behavior, response to dietary excess, insulin secretion, response to glucose, and circadian rhythm. A small population of Limhomedomain transcription factor 1 (Lhx1) neurons was also expanded in the mHFD group compared to the fHFD group and showed enrichment in pathways of insulin secretion and response to glucose. We identified a large neuronal subpopulation marked by Nr5a1 and found the number of these neurons was increased in the mLFD group compared with fLFD group, as well as in fHFD groups compared with fLFD groups. GO pathway enrichment analysis of Nr5a1 specific transcriptomic signatures exhibited enrichment in the feeding behavior pathway, consistent with the fact that Nr5a1 neurons in VMH region can sense glucose levels and conduct insulin and leptin signaling in energy expenditure and glucose homeostasis, with minor feeding control[71]. DEG analysis between maternal dietary groups in neuronal subpopulations also showed that Nr5a1 hits one of the top 3 DEG hubs in neurons (222 DEGs across all the comparisons) (Supplementary Data 4). Together, the single nucleus atlas we generated illustrates the complex neuronal heterogeneities of the offspring hypothalamus under different maternal dietary fat exposures during lactation.

### Heterogeneity of key neurons that regulate food intake

AgRP and POMC neurons are well known to be associated with food intake and energy homeostasis. We subclustered the AgRP neurons based on their transcriptional similarities and found three distinct subpopulations: AgRP-1 (Agrp$^{high}$/Npy$^{high}$), AgRP-2 (Agrp/Npy), and AgRP-3 (Agrp$^{low}$/Npy$^{low}$) (Fig. 4a, b, Supplementary Fig. 4a, b). DAseq and fraction analyses showed there were no abundance differences in the cell proportions of all the AgRP subclusters between offspring sexes and diet groups (Fig. 4c). We have further quantified the percentage of cells expressing Agrp and Npy and their expression in the four offspring groups and there was no significant difference between groups. Lepr was co-expressed in AgRP neurons in our dataset (Supplementary Fig. 4c), but not all the AgRP subpopulations co-expressed Lepr, suggesting AgRP heterogeneity is possibly driven by leptin. AgRP neurons could integrate the action of leptin to regulate both energy

balance and glucose homeostasis[72,73]. We also compared the Agrp expression across four groups and found no significant difference (Fig. 4d). We further quantified the AgRP/POMC ratio and found a higher ratio in the mHFD group (2.69) compared to other groups (fHFD = 2.22, fLFD = 1.91, mLFD = 2.08 respectively). The long noncoding RNA Xist, is widely considered as the master regulator of X inactivation since Xist is part of the X inactivation center (XICC), which harbors additional non-coding RNA genes involved in the same process. It has been recently found as a marker of neuronal aging in female mammal hypothalamus[50]. We found Xist was one of the top upregulated genes in female offspring raised by mothers fed with HFD in all the AgRP neuronal subpopulations in our dataset (Fig. 4d). We also found Brd1 was upregulated in the offspring from maternal HFD groups in both sexes (fHFD vs fLFD: $P_{adj}$ = 0.002, mHFD vs mLFD: $P_{adj}$ < 0.001) and notably, there were more cells expressing Brd1 in the mHFD group with a higher expression level, revealing the hypothalamic adaptation to maternal HFD (Fig. 4d). Previous studies suggested the roles of HDACs as key epigenetic players in obesity development and we observed a higher expression of Hdac3 per cell as well as a higher number of cells expressing Hdac3 in mLFD group ($P_{adj}$ < 0.001, mHFD vs mLFD) (Fig. 4d). We also subclustered the POMC into three subpopulations (POMC-1, POMC-2, POMC-3) by different expression levels of Pomc (Supplementary Fig. 4e–h). Brd1 was significantly upregulated in mHFD group ($P_{adj}$ = 0.022, mHFD vs mLFD).

Avp/Rorb neurons were uniquely expanded in mHFD group (Fig. 3e) and we subclustered the Avp/Rorb neurons into two subpopulations (Avp/Rorb-1 and Avp/Rorb-2, Fig. 4e, f). The Avp/Rorb-1 cluster had higher expression of Dlk1, Vgf and Gnas genes, suggesting a more proliferated cellular status. We further analyzed the differentially expressed genes in the Avp/Rorb clusters and found both Avp and Rorb expression were higher in mHFD group compared with fHFD group (Fig. 4g). We also found Igfbp5, the insulin like growth factor-binding protein 5, was upregulated specifically in the mHFD group compared to fHFD group ($P_{adj}$ < 0.001, mHFD vs fHFD). Gm42418 was highly expressed in male offspring under maternal HFD ($P_{adj}$ < 0.001, mHFD vs mLFD) and this is consistent with previous studies suggesting Gm42418 is involved in the response to HFD[74]. Top/bottom 5 DEGs from the key neuronal populations (AgRP, Avp/Rorb and POMC) were summarized in Supplementary Fig. 4i and Supplementary Data 4.

### Key non-neuronal populations

Over 30% of the cells in the P15 hypothalamus were non-neuronal. The astrocytes were the second-largest population in our atlas, consistent with previous findings in the adult hypothalamus[59]. We identified two distinct astrocyte clusters on the basis of transcripomic similarities: Astrocyte-1 (Gfap) and Astrocyte-2 (Slc7a10) (Fig. 5a, Supplementary Fig. 5a, Supplementary Data 5). We didn't observe an enrichment difference between these two clusters of astrocytes in offspring raised by mothers fed with different diets by DAseq analysis. We further explored the variation among different sub-clusters through pseudotime ordering of nuclei using Monocle3 and identified a trajectory from the Astrocyte-1 to the Astrocyte-2 cluster (Fig. 5b). We further compared Mulan's I test modules and visualized the trajectory related gene expressions along the pseudotime from the Gfap cluster to the Slc7a10 cluster. We found genes including Myoc, Id1, Id3 and Id4 that represents a more-stemmed status were pseudo-gradually downregulated whereas genes including Plpp3, Cst3 were up-regulated along the trajectory (Fig. 5b).

Recent studies have shown that oligodendrocytes also play important roles in regulating energy balance[75,76]. We further subclustered these cells into five subpopulations referred to as ODC-1 to ODC-5. ODC-1 was annotated as oligodendrocyte precursor cells with higher expression of the progenitor marker Pdgfra. The rest of the subpopulations were more differentiated ODCs with Plp1 expression

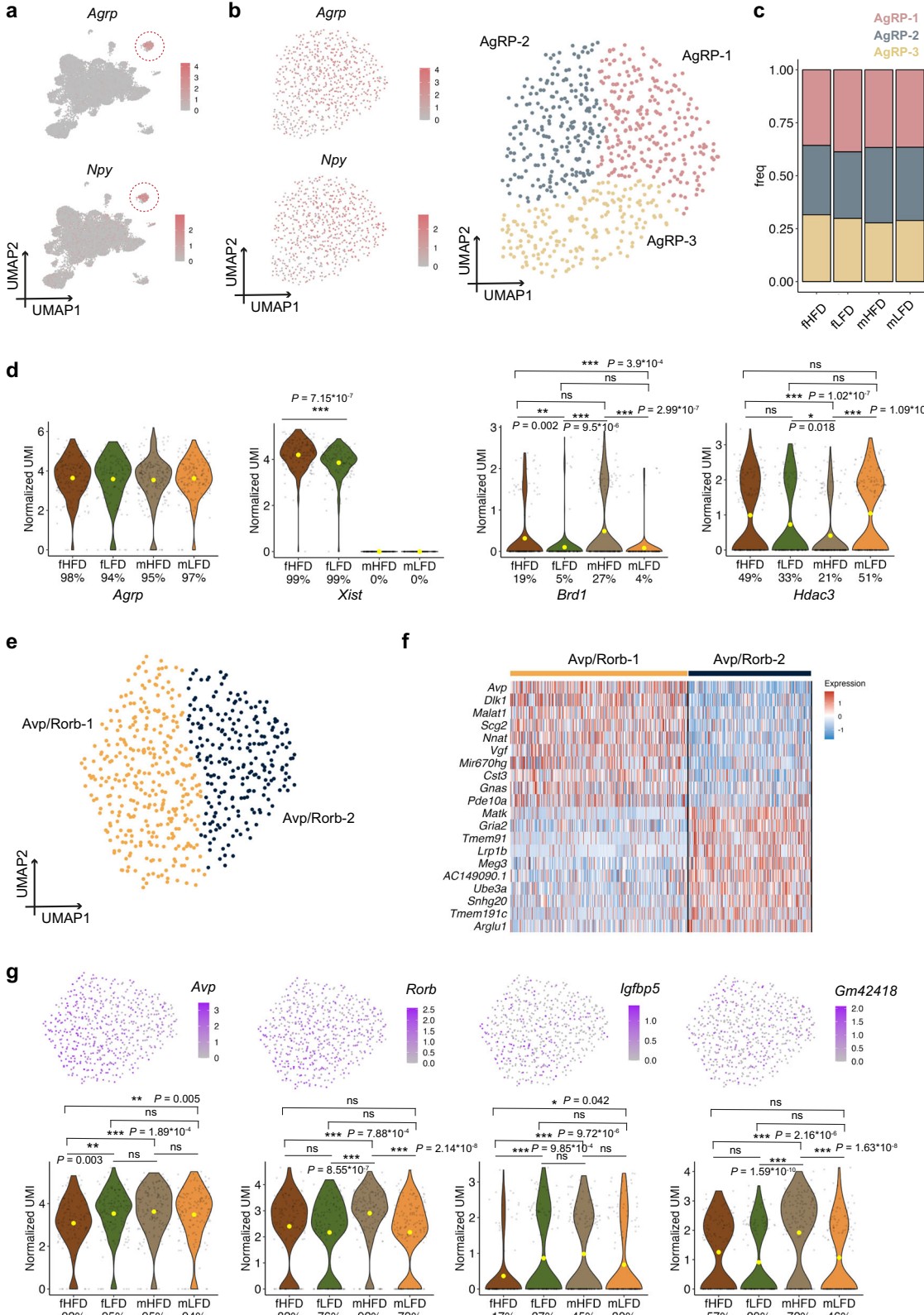

**Fig. 4 | Heterogeneity of AgRP/Npy and Avp/Rorb neurons in P15 offspring hypothalamus. a** UMAP of canonical marker genes for AgRP neurons (*Agrp* and *Npy*) in neuronal subpopulations. **b** UMAP of canonical marker genes for AgRP neurons (*Agrp* and *Npy*) on the extracted AgRP/Npy neurons. Unsupervised clustering of extracted AgRP/Npy subclusters identified three distinct subtypes. **c** Barplot of the cell proportions of AgRP subclusters from fHFD, fLFD, mHFD and mLFD groups. **d** Violin plots of the expression of *Agrp*, *Xist*, *Brd1* and *Hdac3* genes in

four groups in AgRP/Npy neurons. * represents *P* < 0.05 for Wilcoxon test. **e** Unsupervised clustering of extracted Avp/Rorb subclusters identified two distinct subtypes. **f** Heatmap of the top 10 differentially expressed genes in each Avp/Rorb subcluster. **g** Umap and Violin plots of the expressions of *Avp*, *Rorb*, *Igfbp5* and *Gm42418* genes in four groups in Avp/Rorb neurons. * represents *p* < 0.05, ** represents *p* < 0.01 and *** represents *p* < 0.001 for Wilcoxon test. ns represents no significance between comparisons.

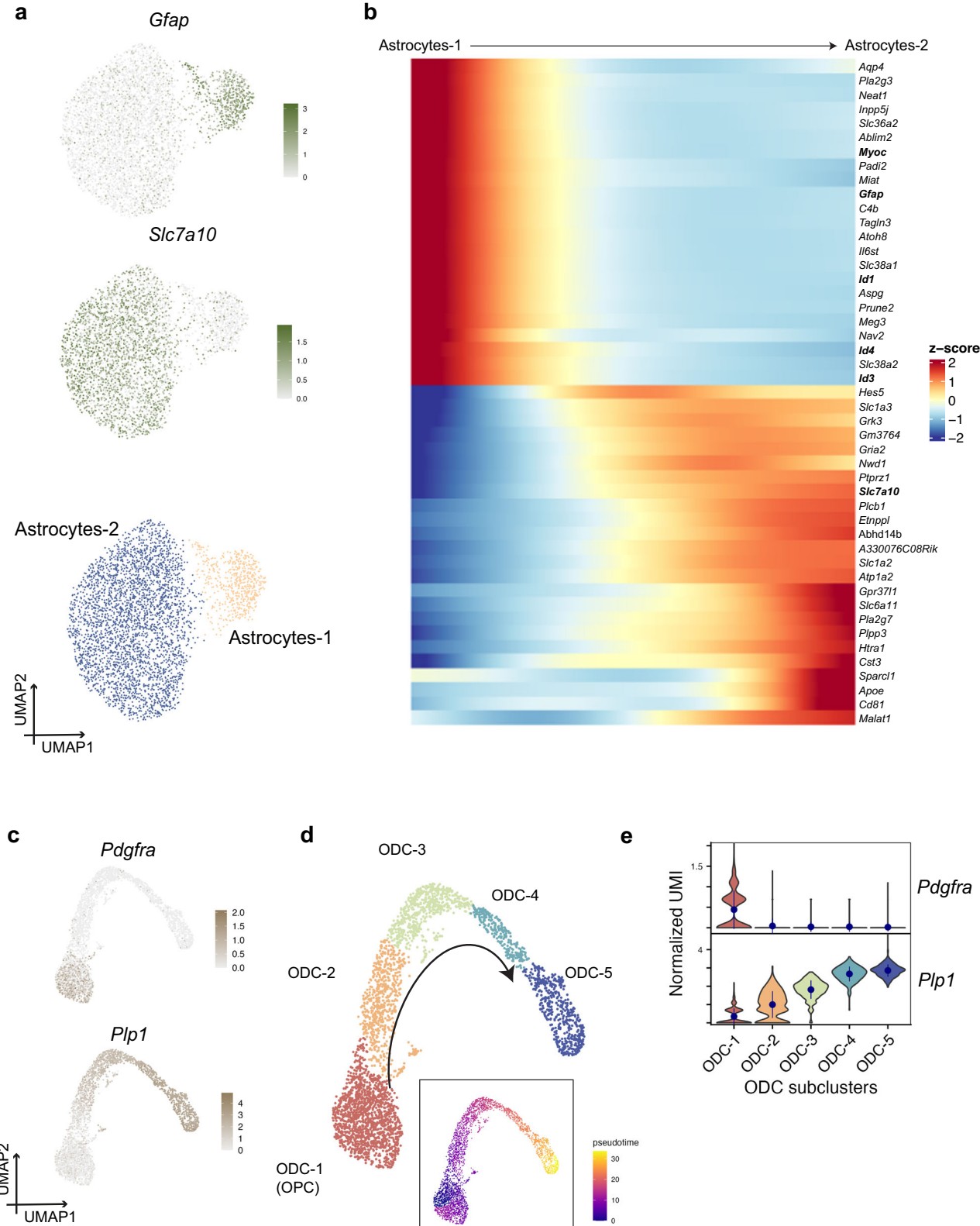

**Fig. 5 | Heterogeneity of astrocytes and oligodendrocytes in P15 offspring hypothalamus. a** Unsupervised clustering of extracted astrocytes identified two distinct subtypes and the expression of canonical marker genes *Gfap* (Astrocytes-1) and *Slc7a10* (Astrocytes-2) on UMAP. **b** Trajectory analysis of astrocytes identified a trajectory from subtype Astrocyte-1 to Astrocyte-2. Trajectory related gene expressions along the pseudotime including *Myoc, Id1, Id3* and *Id4* that represents a more-stemmed status were pseudo-gradually down-regulated whearas genes including *Plpp3, Cst3* were up-regulated along the trajectory. **c, d** UMAP of unsupervised clustering of extracted oligodendrocytes identified five distinct subtypes and the expression of canonical marker genes *Pdgfra* (oligodendrocyte precursor cells) and *Plp1* (oligodendrocytes). Trajectory analysis shows the trajectory from ODC-1 to ODC-5. **e** Violin plots of canonical marker genes *Pdgfra* and *Plp1* in five oligodendrocyte subclusters. *Plp1* shows a gradual change of expression along the differentiation trajectory.

(Supplementary Data 5). We observed a gradual change of *Plp1* along the differentiation trajectory (Fig. 5c–e). Tanycytes are hypothalamic-specific cells and we further subclusted them into ependymal cells (*Dnah12*, *Ccdc153*), tanycytes alpha+beta1 (*Fzrb*, *Vcan*) and beta2 sub-populations (*Col24a1*, *Scn7a*) (Supplementary Fig. 5b, c, Supplementary Data 5). We did not observe significant changes of the cellular proportions in ependymal/tanycytic subclusters between different maternal dietary groups.

### Cell-cell communications in offspring hypothalamus revealed sex- and maternal diet-specific effects

To understand the cell-cell interactions between cell populations, we first applied CellChat to the nine key lineages. We found that neuron populations were the hotspot of cellular interactions in all four groups. Both female and male offspring from maternal HFD groups had significantly enriched cell-cell interactions compared with those from maternal LFD groups. Because neurons exhibited the most intensive cellular interactions among all the cell types, we further performed cell-cell interaction analysis on neuronal subclusters with non neuronal clusters. We identified 10,595 cell-cell interactions in mHFD groups and 9949 in mLFD, 8120 in fHFD and 7050 in fLFD groups (Fig. 6a). There were more interactions in the offspring raised by maternal HFD compared with those by maternal LFD.

We then focused on the interactions between food intake regulator AgRP/Npy neurons and astrocytes (top scoring non-neuronal populations) as the interaction weight was the strongest across key metabolic relevant clusters (Fig. 6b, Supplementary Fig. 6a–c). We validated the differential interaction strengths using immunohistochemistry staining. Using a Npy-hrGFP mouse model exposed to the same maternal diets, we stained astrocytes on postnatal day 15 in male and female offspring brains using the canonical marker S100β. We found Npy-GFP positive cells were largely enriched in the hypothalamic ARC region. Some Npy neurons marked by GFP also co-expressed S100β, which indicated that Npy neurons and astrocytes were co-localized in the ARC area. To further determine the effect of maternal dietary fat on the co-localization of Npy neurons and astrocytes in both male and female offspring, we quantified the amount of Npy/astrocyte co-localized cells in the ARC region for all four groups. The co-localization of cells in male offspring raised by mothers fed on HFD was significantly higher than the male offspring raised by mothers feeding on LFD ($t$ test, $F = 0.346$, $P = 0.046$). However, there is no significant difference between the co-localization of cells from female offspring raised by mothers fed with HFD during lactation and those female offspring raised by mothers fed the LFD ($t$ test, $F = 0.32$, $P = 0.408$). (Fig. 6c, d). No significant differences ($t$ test, fHFD vs fLFD: $F = 0.155$, $P = 0.764$; mHFD vs mLFD: $F = 0.324$, $P = 0.0826$) were observed between groups after normalization (computed by the ratio of the cells co-localized with Npy neurons and astrocytes to the total number of cells stained by DAPI) (Supplementary Fig. 6j). The same trends were shared using both the number of co-localized cells as well as the ratio of co-localized cells to total cells in four groups, indicating that increased cellular interactions might be occurring between AgRP/Npy neurons and astrocytes in male and female offspring raised by mothers fed HFD during lactation and the response to maternal effect in a sexually dimorphic manner, coherent with the predictions using CellChat.

Notably, we observed that male offspring under maternal HFD exhibited highly enriched interactions of NEGR (neuronal growth regulator) pathway (Negr1-Negr1 interaction pair) within AgRP neurons in male offspring raised by mothers fed with HFD (Fig. 6e–g, Supplementary Fig. 6d–h, Supplementary Data 6), in comparison with male offspring under maternal LFD or female offspring under maternal HFD. Furthermore, we found an enrichment of NCAM1 (Ncam1-Ncam2/1) pathway interactions between AgRP neurons and astrocytes in male offspring under maternal HFD.

## Discussion

Studies in both rodents[6,8,14,19,77,78] and humans[17] have shown that high maternal dietary fat intake during pregnancy and/or lactation increases the susceptibility of offspring to later obesity, combined with insulin and leptin resistance during adulthood. In this study, we combined quantification of the metabolic traits from mothers and offspring C57BL/6 mice under low and high fat feeding during lactation, with a single nucleus transcriptomic atlas of the female and male offspring hypothalamus at P15 (peak lactation) to understand the cellular functional states and transcriptomic alterations in response to the maternal dietary fat pertubation.

Multiple rodent studies have shown that elevated maternal dietary fat content increased offspring BW[18,79,80]. Maternal HFD during lactation increased adiposity of early postnatal offspring mice and this maternal effect exacerbates offspring obesity and related metabolic syndrome in males, but less so in females, in later life[15,18,21,23–25]. Consistent with the previous results in different mouse strains, our study showed that offspring raised by mothers fed HFD during lactation had higher body weight, fat and lean mass than those mothers were fed LFD.

The hypothalamus is a major brain region for energy homeostasis and has been shown to be a critical target of maternal HFD[8,42,81,82]. To understand why males are more sensitive to the maternal HFD exposure than females in adulthood, and how the early postnatal hypothalamic transcriptomic landscape is altered in response to the maternal diet is an important goal. The diversity of neuronal and non-neuronal cell subpopulations and states has enabled the identification of the cellular heterogeneity of rare and/or transitory cell types and statuses. We captured all the major cell types that were observed in other single cell atlases of adult murine hypothalamus[44,46–55,57,58] and identified 30 neuronal subpopulations across the different regions in the hypothalamus. Although the key cell populations were similar between maternal HFD and LFD groups, there was an expansion in the numbers of Hdc, AgRP and Avp/Rorb neurons in mHFD groups when compared with fHFD and/or mLFD groups. Histamine (HA) is a monoaminergic neurotransmitter synthesized from L-histidine through histidine decarboxylase (HDC)[83]. Hypothalamic neuronal HA has also been shown to regulate feeding behavior and energy metabolism as a target of leptin action in the brain[84,85]. Hdc neurons are activated by fasting[34], and a small population of Hdc neurons is sensitive to insulin-induced hypoglycemia[86]. Hypothalamic AgRP and POMC are 'yin/yang' master regulators that promote/inhibit food intake, previous studies observed a higher ratio of AgRP/POMC expressing neurons in offspring raised by mothers with maternal overnutrition during pregnancy and/or lactation[5,8,40,41]. AVP neurons in the PVN region acutely inhibit food intake in mice[87]. Retinoic acid receptor-related orphan receptor-β (RORβ) together with RORα and/or RORγ regulate the transcription of *Clock* and *Bmal1* in the hypothalamic SCN[88]. Whereas the roles of AVP/Rorb neurons have not been addressed previously. Lhx1 is essential for terminal differentiation and function of the hypothalamic SCN, the key regulator of light-entrained cicadian rhythms that spontaneously synchronize circadian clocks[89,90]. These neurons are functionally categorized to glucose and energy homeostasis, insulin signaling, feeding behavior and circadian rhythm related cellular pathways by our GO enrichment pathway analyses. The expanding of these neurons in mHFD groups indicates that male offspring are more sensitive in response to the maternal programming when exposed under the same maternal HFD during lactation by changing the proportions of multiple neuronal subpopulations, which might contribute to the increased susceptibility to adiposity in later life.

Hypothalamic astrocytes are particularly affected by high caloric diets. Astrocytes located in the ARC altered morphologically in response to a high caloric diet, affecting their physical interactions with neurons and blood vessels in mice[91], and exhibited distinctive

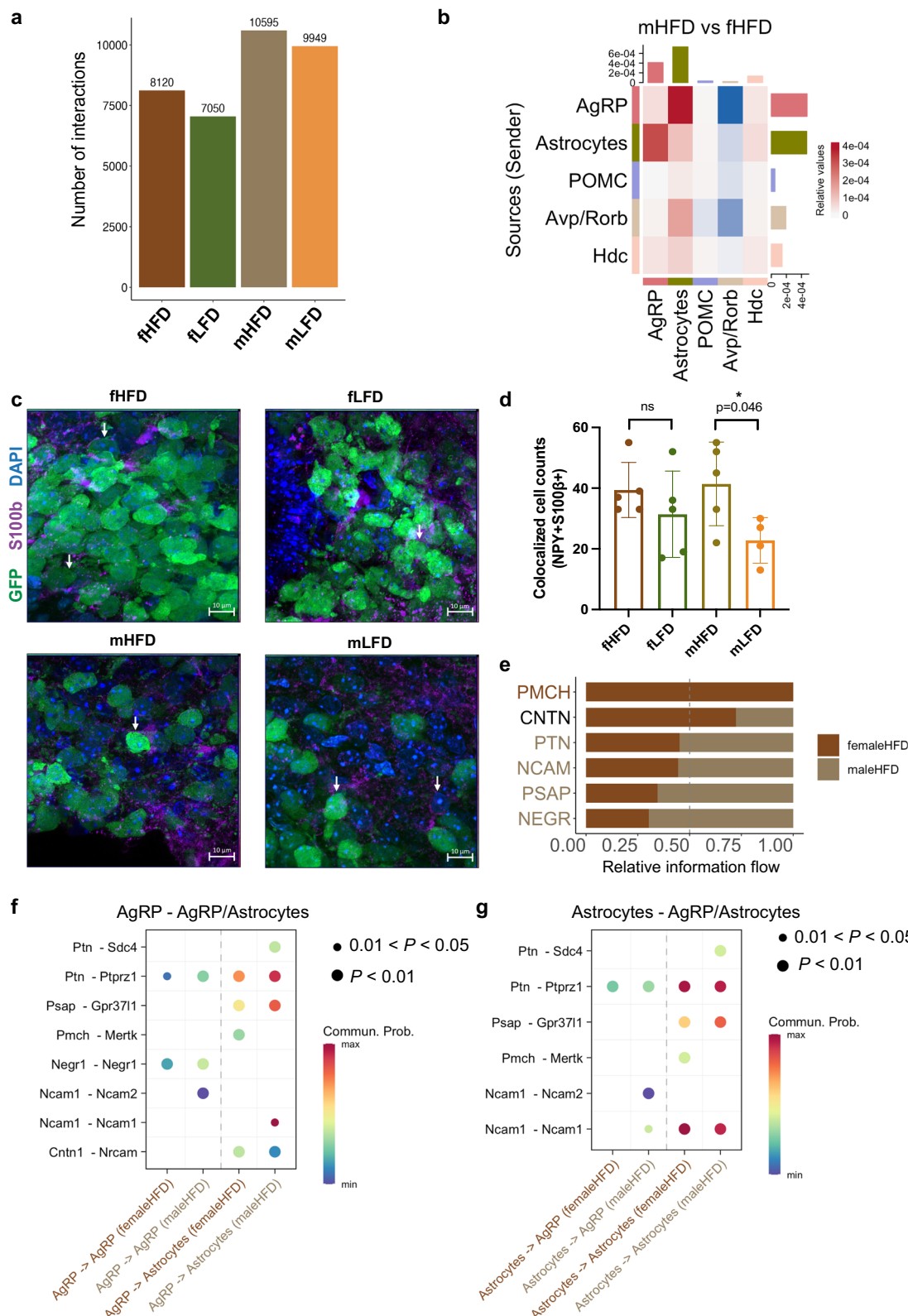

**Fig. 6 | Cell-cell interactions in offspring hypothalamus revealed sex- and maternal diet-specific effects. a** Total numbers of cell-cell interactions in fHFD, fLFD, mHFD and mLFD groups. **b** Overall interactive signaling changes on key neuronal subpopulations and top scoring non-neuronal populations between mHFD and fHFD groups. **c** Immunohistochemistry staining shows the co-localization of AgRP/Npy neurons and astrocytes. Green cells marked by GFP represent AgRP/Npy neurons, purple cells marked by S100β represent astrocytes, blue dots marked by DAPI represent the nuclei. White overlapping by both GFP and S100β represents the colocalization of AgRP/Npy neurons and astrocytes. **d** Cell

counts of colocalized AgRP/Npy neurons and astrocytes in four groups. Biologically independent animals for fHFD group were 5, for fLFD group were 5, for mHFD group were 5, for mLFD group were 4. *p* value by two-sided Student's *t* test. Values are means ± s.d. * represents *p* < 0.05 for Wilcoxon test. **e** The identified key pathways of the interactive signaling between mHFD and fHFD groups. **f, g** The identified key up-regulated and down-regulated signaling ligand-receptor pairs between the interactions of AgRP neurons and astrocytes in mHFD and fHFD groups. Source data is provided as a Source Data file.

temporal high caloric diet-induced transcriptomic modifications[92]. We observed increased cell-cell interactions between AgRP/Npy neurons and astrocytes in the offspring raised by mothers exposed to HFD exposure. The altered physical interactions of astrocytes-AgRP/Npy neurons together with expanded AgRP/Npy populations in male off-spring under maternal HFD programming might be important in increasing their susceptibility for developing obesity at adulthood. Genome-wide association studies (GWAS) in large cohorts have identified about more than two thousand loci linked to body mass index (BMI), a trait that is often used as a proxy for obesity[93–95]. The majority of the variants identified by GWAS are enriched in neurodevelopment[96–98]. *NEGR1* is one of the top genes associated with BMI by GWAS[27,99,100]. NEGR1 is a cell surface molecule extensively found in the brain, and involved centrally in synaptogenesis, neurite outgrowth and cell-cell adhesion[101], also plays a role in brain connectivity[102,103], an important process in obesity[104]. Through cell-cell interaction analysis, we observed that outgoing and incoming signals of the NEGR pathway were highly enriched in obesity relevant cell subpopulations such as Lhx1, Nr5a1 and Sst in offspring from maternal HFD groups which further highlighted the higher chance of developing obesity and related metabolic disorders in offspring raised by mothers fed HFD.

In summary, our study reveals the key lineages of the postnatal hypothalamus at peak lactation under maternal dietary fat exposure during lactation and links together alterations in the metabolic phenotype of male and female offspring with the corresponding hypothalamic cellular subpopulation expansions and cell-cell interactions. We observed transcriptional variations across cell subpopulations among different maternal dietary fat groups, which were consistent with male offspring being more sensitive to maternal diet. These results revealed the sex-specific effect in response to maternal overnutrition at early life at single cell resolution. We further built a web portal (https://mouse10x.shinyapps.io/p15atlas/) to enable the users to explore the genes of interest, which would be of great value to the scientific community concerned with maternal obesity or maternal overnutrition and their effects on early life development in offspring. It would also expand the knowledge of existing resources of murine hypothalamic single cell atlas. Although we observed a sexually dimorphic response of offspring to maternal high fat diet combined with physiological and single nucleus transcriptomics analyses, further studies are needed to include a wider time window including puberty, adult and aging as well as a larger sample size of individuals to address the detailed mechanisms of when and how the male and female offspring diverge in response to maternal over-nutrition at the single cell level and the effect on the adulthood. Additionally, the experimental validation for the expansion of other key neuronal subpopulations such as Avp/Rorb and Hdc is needed in separate studies. Furthermore, in this study we only fed the mothers diets with two fat contents (45% and 10%), more dedicated dietary design (such as dietary fat composition, low fat high sucrose diet etc) is required in the future to address the specific components in the maternal diet affecting sex dimorphic manner.

## Methods

### Ethical statement

All animal procedures were reviewed and approved by the animal ethical panel at Institute of Genetics and Developmental Biology, Chinese Academy of Sciences (approval number: AP2020001).

### Mice

C57BL/6 N mice were purchased at 5 weeks old from Charles Rivers, Beijing and acclimated to the SPF facility before starting the baseline. Npy-hrGFP (#006417) with a *C57BL/6J* background was obtained from Jackson Laboratory and maintained at the facility afterwards. Primers and PCR conditions for genotyping were followed by the protocol of

the Jackson lab. All animals were kept at $23 \pm 1\,°C$ with a dark–light cycle of 12 h–12 h (lights on at 0730 h). Mice were fed a standard low fat chow diet [crude fat ≥4% by weight, crude protein ≥20% by weight (Huafukang Bioscience, Beijing, China)] before the baseline. Female mice at 11 weeks of age were fed with LFD (10% fat by energy, D12450B, Research Diets, New Brunswick, NJ, USA) for 2 weeks as the baseline. Mice were mated at 13 weeks old and remained on LFD until parturition. Female mice were then randomly allocated to LFD and HFD (45% fat by energy, D12451, Research Diets, New Brunswick, USA) on lactation day 1 (Fig. 1a). Litter sizes were manipulated to 6 at lactation day 1 for all dams and recorded daily during lactation. Female mice occasionally cull pups during lactation. We retained litters with a litter size over 5. Mice were sacrificed by $CO_2$ overdose before sample collection as well as at the end of experiment. Final sample sizes of C57BL/6 N mice allocated to maternal LFD and HFD were 9 and 6 for metabolic traits measurements. Final sample sizes of Npy-hrGFP mice offspring included in immunohistochemistry validation from fHFD group were 5, from fLFD group were 5, from mHFD group were 5, from mLFD group were 4.

### Metabolic phenotype measurements

Maternal BW and FI were measured daily during the whole experimental period. Litter mass and pup mass were measured daily during lactation. Total in vivo body fat and lean content of the females were evaluated by magnetic resonance spectroscopy (EchoMRI, Houston, TX, USA) on the day before mating, lactation days 1, 10 and 16. The total in vivo body fat and lean content of the litters were also measured at lactation day 16. Feces produced by female mice during lactating days 13–15 were collected, separated from the bedding manually, and oven-dried at 60 °C to a constant mass (14 days). Samples of each diet were also weighed and dried to a constant mass to obtain dry mass. The water content of the diets was measured to correct the FI. The calorific values of feces and diets were determined by a Parr 6400 Calorimeter (Parr Instrument Company, Moline, IL, USA). Metabolizable energy intake (MEI) was calculated as below[79]:

$$MEI = (M_{food} \times GE_{food}) - (M_{feces} \times GE_{feces})$$

where $M_{food}$ is the dry mass of FI in g day$^{-1}$, $M_{feces}$ is the dry mass of feces produced in g day$^{-1}$, $GE_{Food}$ is the gross energy content of the food (KJ g$^{-1}$) and $GE_{feces}$ is the gross energy lost in feces (KJ g$^{-1}$).

The doubly labeled water (DLW) method[105] was used to measure daily energy expenditure (DEE) from the elimination rates of $^2H$ (deuterium) and $^{18}O$ in lactating females during peak lactation (day 13–15). Measurements of DEE were made to determine the milk energy output (MEO) from the difference between MEI and DEE[106]. Individual mice were weighed to ±0.01 g using a balance (BSA2202S; Sartorius, Göttingen, Germany) and labeled with an intraperitoneal injection of approximately 0.1 g of water containing enriched $^2H$ (36.3 atoms%) and $^{18}O$ (59.9 atoms%). Syringes used to inject the DLW were weighed (±0.001 g; JA2003N; Hangping, Shanghai, China) immediately before and after the injection to provide an accurate measurement of the amount of isotope injected. Mice were placed in their cages during the 1 h equilibration period. An initial 30–80 μl blood sample was collected from the orbit 1 h after the injection[107]. Blood samples were immediately flame-sealed into pre-calibrated 100 μl capillaries. A final blood sample was collected 48 h after the initial blood sample to estimate isotope elimination rates. Samples of blood in capillaries were vacuum-distilled[108]. A liquid water analyzer (Los Gatos Research, Mountain View, CA, USA) was used to analyze the isotope ratios of $^{18}O{:}^{16}O$ and $^2H{:}^1H$. The samples were run alongside a range of international and in-house standards that were used to correct the raw data for daily machine variation. For each lactating mouse, initial $^2H$ and $^{18}O$ dilution spaces were calculated by the intercept method and then converted to mass, assuming a molecular mass of body water of 18.02 and

expressed as a percentage of body mass before injection. The intercept method was used to estimate the body water pool as this gives the best estimate compared with desiccation[109]. The final $^2H$ and $^{18}O$ dilution spaces were inferred from the final body mass, assuming the same percentage of body mass as measured for the initial dilution space. For calculation of DEE based on $CO_2$ production, single pool model[110] was used as recommended for small mammals[111]. Energy equivalents of rates of $CO_2$ production were calculated using a conversion factor of 24.03 J $ml^{-1}$ $CO_2$, derived from the Weir equation[112]. Mice were scarified at lactation day 16, one male pup and one female pup from each litter were dissected. The brain, brown adipose tissue (BAT), subcutaneous fat (SUB), mesenteric fat (MWAT), heart, liver, lungs, kidneys, pancreas, stomach, spleen, small intestine, caecum, colon and testes (only for male pups) were immediately dissected and weighed on a ± 0.001 g balance (JA2003N; Hangping).

### Single nucleus isolation

The dissociation protocol was revised based on the 10X Sample Prep Demonstrated Protocol (CG000124, Rev D). Briefly, male and female pup brains at P15 were rapidly dissected between 9 and 12 am and the hypothalamus was spooned out immediately and cut into tiny pieces in HEB buffer (Hibernate A®, 2% B27®, 0.5 mM GlutaMAX™). Hypothalamus pieces were then incubated in 2 ml pre-chilled lysis buffer (10 mM Tris-HCl, 10 mM NaCl, 3 mM $MgCl_2$, and 0.1% Nonidet™ P40 Substitute in Nuclease-Free Water, 0.1 U/μl RNase Inhibitor) on ice for 15 min. 2 ml of HEB buffer were added to the lysed tissue and triturated the tissue with a fire polished silanized Pasteur pipette 10–15 times. After trituration, a 30 μm MACS® SmartStrainer was used to remove cell debris and large clumps. The single nucleus suspension was centrifuged at 400 g for 5 min at 4 °C and washed in a wash and resuspension buffer (1X PBS with 1.0% BSA and 0.2 U/μl RNase Inhibitor). Subsequently, the dissociated hypothalamic single nuclei were stained with Trypan Blue (1:1) to evaluate the nuclei quality under a microscope. The nuclei suspension concentration was determined by CountStar Rigel S2 (Alit Biotech, Shanghai), once the target nuclei concentration of 300–600 nuclei/μl was obtained, proceed immediately with the 10x Genomics® Single Cell Protocol.

### Single-nucleus RNA-seq library preparation, sequencing and data pre-processing

Single nucleus RNA-seq libraries were constructed using the 10x Genomics® Chromium Single Cell Controller and the next GEM Single Cell 3' Reagent Kit v3 (10x Genomics, USA) according to the user's guide. Nuclei suspensions were loaded on the Chromium Controller to generate single nuclei gel beads in the emulsion (GEM). Captured nuclei were lysed to release mRNA which was subsequently barcoded through reverse transcription of individual GEMs. Using a thermo cycler (Eppendorf 6321 Mastercycler pro, Hamburg, Germany) to reverse transcribe, the GEMs were programmed at 53 °C for 45 min, followed by 85 °C for 5 min, and held at 4 °C. The cDNA library was then generated, amplified, and assessed for quality control using the Agilent 4200. The single nucleus RNA sequencing was further performed on the Illumina Novaseq 6000 sequencer. Raw data were processed with Cell Ranger (version 4.0.0) with default parameters for each sample and mapped to the mm10-3.0.0 genome to generate unique molecular identifier (UMI) expression matrices.

### Quality control and integration of single nucleus transcriptomic data

Nuclei in which <200 or >6000 detected genes, <500 or >30000 UMIs, or with more than 5% reads mapping to mitochondrial genes were removed. SCTransform normalization (v2) in Seurat (4.1.0) was performed separately for each sample. Subsequently, 3000 variable feature genes were selected using the 'SelectIntegrationFeatures' function. PCs were found with variable genes using the 'RunPCA'

function, and the top 40 PCs were used for downstream analysis. We further integrated all the samples using 'PrepSCTIntegration', 'FindIntegrationAnchors' and 'IntegrateData ' functions to integrate individual samples as suggested by Seurat tutorials.

### Clustering and cell type annotation

Uniform Manifold Approximation and Projection (UMAP) analysis was performed to further reduce variation to two dimensions with 'RunUMAP' function. These two UMAP dimensions were used to identify major clusters with the 'FindNeighbors' and 'FindClusters' function at a resolution of 0.1. Cell identities were determined by marker genes generated from 'FindAllMarkers' function as well as the canonical marker genes from published literature. For subclustering, neurons, astrocytes, tanycytes and oligodendrocytes were subsetted and subclustered respectively, AgRP, POMC and Avp/Rorb subclusters were further subsetted on neuron subclusters. All the reclustering was done using the above method with refined resolution.

### Symphony analysis

Symphony is an efficient single cell reference atlas mapping tool[67]. We build a Symphony reference using the hypoMap dataset. We downloaded the hypoMap.rds file and extracted the expression matrix (counts), metadata and UMAP embeddings. For reference building, we chose the 2000 most variable genes and subsetted the dataset with the variable genes, then ran PCA with 20 dimensions and calculated the UMAP. To make use of the UMAP in original Seurat object and the query dataset, we substituted the UMAP embeddings calculated by Symphony with the one stored in the original Seurat object with the function 'buildReference_on_UMAP'. For query mapping, we extracted the expression matrix (counts), metadata from the query dataset, then we mapped the query dataset on the reference dataset using the 'mapQuery' function in Symphony.

### Differential abundance analysis (DAseq)

DAseq was adapted to compare the composition enrichment on the full dataset as well as the neurons. Briefly, we extracted the meta.data from the Seurat object and did a pair-wise comparison between fHFD group with mHFD group, fHFD group with fLFD group, fLFD with mLFD group and mHFD with mLFD group by including the replicates information in labels.

### Differential expressed genes (DEGs) analysis

To run differential gene expression analysis, we used normalized counts that are stored in the data slot of the SCT assay. Prior to performing differential expression, we first ran 'PrepSCTFindMarker' to set the fixed value, followed by 'FindMarkers' function by "MAST" method (assay = "SCT", test.use = "MAST", logfc.threshold = 0, min.pct = 0, min.cells.feature = 1) to identify differentially expressed genes between the comparisons of mHFD vs mLFD, fHFD vs fLFD, mLFD vs fLFD and mHFD vs fHFD groups including the replicates. Genes with an adjusted $P$ value < 0.05 were considered as significant DEGs between groups based on previous publications. Marker genes of each cell population were determined by FindAllMarkers function (logfc.threshold = 0.25, assay = "SCT") with adjusted $P$ value < 0.05.

### Pathway enrichment analysis

Marker genes of each major cluster that identified by FindAllMarkers function described above were used as the input to perform GO pathway enrichment analysis using 'enrichGO' function from Cluster-Prolifier (version 4.9.1[113]). All the significant DEGs between different comparisons (mHFD vs mLFD, fHFD vs fLFD, mLFD vs fLFD and mHFD vs fHFD) were also extracted by 'Findmarker' function described above, significant DEGs were then fed to 'enrichGO' for GO[114] enrichment analysis.

## Trajectory analysis

Monocle 3 was used to run the trajectory analysis on the astrocyte and oligodendrocyte clusters. We first extracted the meta.data and expression counts from Seurat object to build a monocle3 cds object. We further substitute the umap embeddings and cluster information in the monocle3 cds object with integrated Seurat object. We used 'learn-graph' for the trajectory analysis and 'plot_cells' to plot the trajectory.

## Cell-cell interaction analysis

Cellchat was used to profile the cellular interactions. We first performed the cellular interaction analyses on the full dataset on the key lineages and then included all the neuron subclusters on four groups separately and then compared the cell-cell interactions.

## Immunohistochemistry

Female and male NPY-hrGFP mice offspring raised by mothers fed HFD and LFD during lactation were sacrificed on postnatal day15. Fresh brain tissues were dissected and proceeded with perfusion. Brain samples were immersed with 4% PFA overnight and treated with 30% sucrose solution at 4 °C for 48 h. Samples were then embedded with OCT and serially sliced with a thickness of 30 μm and stored at −20 °C for further use. For staining, brain sections were incubated with solution containing 5% BSA (Bovine Serum Abumin) and 0.3% Triton for 1 h. S100 beta rabbit monoclonal antibody was used as primary antibodies (AF1945, Beyotime) (1:500) at 4 °C for 8−12 h. Goat anti-rabbit IgG antibody (Alexa Fluor® 647, ab150087, Abcam) (1:1000) was used as secondary antibody by incubating the slides at room temperature for 1 h in the dark. Nuclei were counter-stained with Dapiprazole hydrochloride (DAPI) (D9542, Sigma-Aldrich) (1:1000). After PBST washing for 3 times, the slides were sealed with mounting solution. The fluorescence images were acquired using confocal microscope (LSM980, Zeiss). Imaris (version 9.8.0) was applied for the cell counting and co-localization analysis. It was used to locate different fluorescent labeled cells in the ARC area of hypothalamus, all fluorescent signals were then automatically adjusted by the threshold and manually labeled. We then set up a certain area to normalize cell counting between different slides. After the generation of spots, the cell numbers at different channels were calculated, co-localization was determined by setting the distance threshold of spots (10 μm).

## Statistical analysis

BW and FI differences during baseline were measured using ANONA. Repeated measures general linear models (RM GLM) during pregnancy and lactation. Day is a repeated factor. Differences in BW, FI, litter/pup mass and body composition during experiments were tested using RM GLM with diet as the fixed factor, and day as the repeated factor. Body fat and lean content of both mothers and weaned offspring were tested using ANOVA with diet as a fixed factor. Changes in MEI, DEE and MEO between dietary groups were compared using GLM with diet as fixed factor and BW as a covariate[115], interaction between the fixed factor and the covariate was also tested. Organ morphology changes between dietary groups were also conducted using GLM with diet as fixed factor and BW as a covariate. If the result showed no significant effects while including the interaction or covariate effect, the significance analysis of the fixed factor was analysed individually. If found, the effects by the interaction or covariate would be taken into consideration. Imaging data were tested using $t$ test. Data are represented as means ± s.d. All data were tested for normality prior to analysis. All statistical analyses were performed using IBM SPSS Statistics for Macintosh (version 24).

## Reporting summary

Further information on research design is available in the Nature Portfolio Reporting Summary linked to this article.

## Data availability

The raw mouse hypothalamus snRNA-seq FASTQ files generated in this study have been deposited in the Gene Expression Omnibus (GEO) database under accession code GSE217677. The reference genome used for raw sequencing reads alignment is mm10-3.0.0. The processed mouse hypothalamus data is also available at GSE217677. Dataset related to HypoMap project used in the study are available at: https://www.repository.cam.ac.uk/items/8f9c3683-29fd-44f3-aad5-7acf5e963a75). An interactive data portal is developed based on the processed dataset and is available online https://mouse10x.shinyapps.io/p15atlas/. All experimental data supporting the findings of this study are available within the paper and its Supplementary Information and Source data are provided with this paper.

## Code availability

All scripts used in the current study is available on GitHub (https://github.com/hyhy200g/MouseHypothalamus_snRNA-seq) and on Zenodo (https://doi.org/10.5281/zenodo.10654924)[116], the used packages are listed in the Methods session.

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

## Acknowledgements
This study was funded by the National Key Research and Development Program of China (2018YFA0800900) to J.R.S. and National Natural Science Foundation of China (32371195) to G.W. and a new PI Start up grant of Fudan University (JIH2303132Y) to G.W. We would like to thank Dr. Teresa Valencak for shipping the blood samples to the UK during the pandemic, we also would like to thank Marina Samatiou for her technical assistance with the DLW measurements. We are grateful to Dr. Min Dai for his assistance and suggestions on the snRNA-seq data analysis. We thank the staff of the animal facility for their care of the animals. The computation in this research was supported by the Medical Science Data Center of Fudan University. The snRNAseq diagram in Fig. 2a was adapted and created with BioRender.com (agreement number: KT26GDTQ01).

## Author contributions
Conceptualization, J.R.S. and Y.H.; Methodology, Y.H. and B.L.; Investigation, Y.H., A.W., A.M.R., Z.J. and C.H.; Data analysis, Y.H., G.W., W.Z. and L.Z.; Writing, Y.H., A.W., G.W., B.L., W.Z. and J.R.S.; Supervision, J.R.S. and G.W.

## Competing interests
The authors declare no competing interests.
