## [Peer Review File · Nature Communications]

Maternal dietary fat during lactation shapes single nucleus transcriptomic profile of postnatal offspring hypothalamus in a sexually dimorphic manner in miceREVIEWER COMMENTS

Reviewer #1 (Remarks to the Author):

The manuscript addresses the interesting topic of maternal programming in the development of metabolic diseases. Previous studies have highlighted the role of maternal high fat diet feeding in mice, specifically during lactation to promote obesity and impaired glucose metabolism in the offspring. Interestingly, male offspring is particularly vulnerable to the adverse effect of maternal HFD feeding in these models. Nevertheless, the underlying neuronal mechanisms underlying these effects remain only partially defined. The authors have employed sn sequencing of hypothalamic tissue in the offspring of mice exposed to maternal HFD feeding during lactation. They report gender specific effects of the intervention with respect to cell population abundance as well as AgRP-neuron/astrocyte interactions. While the topic of the study is interesting and the sn sequencing data appear to be of adequate quality and potentially great value, the limited exploration of this unique resource does not yield the expected mechanistic insights.

Specifically, the major concerns relate to:

1. The phenotypic characterization of animals undergoing the dietary intervention (Fig. 1) is sound and largely represents what has been previously published. However, Figure labelling and the text do not match, which makes it hard to follow the result section.
2. The data represented in Fig 3D do not add much relevant information to the actual question of the work.
3. The approach to identify differential abundance with DA-seq could be a good choice to look into neuron subclusters. However, given the known difficulties with inter-sample variation when analyzing differential abundances of cell types in sc-seq data, the authors should clarify whether replicate information was included in the DA-seq analysis. If not, possibly include it and/or report results with another well-established approach such as scCODA. Furthermore, it would generally be helpful to report the cell type composition (specifically for those cell types that show differences) in cell counts/percent on a replicate level similar to Figure S2C (but for all relevant comparisons and cell type levels). In addition, in Figure S2C it is not clear which replicate comes from which experimental group. Overall, this would help to make the abundance analyses more convincing.
4. Cell type abundance differences should be confirmed by independent approaches, such as in situ hybridizations and quantitative analyses of cell type representation comparing controls and dietary intervention groups.
5. Regarding differential gene expression analyses, the authors implemented a workflow as recommended by Seurat for marker genes of clusters and maybe also for differentially expressed genes per group although this remains unclear from the method description. It would be nice to also include the biological replicate in the MAST model if possible (or state in the methods that it already was included). Furthermore, the authors do not really report the results of a DEG analysis (although it being the very core of what the data can provide besides the DA). Only a handful of genes are shown with violin plots (Fig. 4D). But important questions would include, which cell types are perturbed (i.e., in a simple approach: show strong are the changes (in terms of number, type of DEGs)), and is it possible to look into enriched GO terms for these cell types. In the Figures one could visualize more than 2-3 genes by using heatmaps/volcanoplots of the DEGs of the most relevant cell types (Agrp, POMC, AVP/Rorb, whatever else can be identified through DEG and/or DA analyses). Additionally, the authors should then also more systematically evaluate DEG differences between males and females.
6. Regarding the subclustering provided in Fig. 4, the validity of the subclustering based on the shown marker genes remains unclear. Taking for example POMC, why do the authors not cluster by previously described marker genes such as Ttr, Anxa2, Lepr, Glp1r (Campbell et al.). It would be nice to compare these subclusterings with public data, e.g. via HypoMap to demonstrate that they follow previously identified groupings and then connect that more clearly with the DA and DEG analyses. How does Fig. 4C relate to the previous DA-seq between mHFD and HFD findings? One could highlight those on the AGRP UMAP as well. Are there differences in DEG numbers between subtypes?

7. While it is nice to show the differentiation of Oligos and the differences between GFap+ and Gfap-Astrocytes (Fig. 5), this Figure is not really relevant to the maternal diet question and the different conditions are not considered in this analysis at all, aside from the statement that there were no differences.

8. The authors claim that AGRP and astrocytes have strong interactions, but it is not clear from any plot shown in Fig. 6/S6 how they conclude that. Generally, it would be nice to not only show which signaling molecules drive the interactions but also which cell types interact with each other (and which interactions change). In addition, while the authors employ NPY reporter mice to study potential AgRP-neuron/astrocyte interactions, these sections could also be used to quantify NPY neurons across experimental groups to provide evidence for the postulated changes in AgRP neuron numbers (see above)

9. Finally, the manuscript lacks supporting data with in relation to numerous aspects of the study. Table S3 should also show percentages of cell types etc. per replicate and experimental group. While Table S4 reports the marker genes per group, the actual DEGs (between experimental groups per cluster) should be reported too. There should be a Suppl. table (or source data) for marker genes of the subclustering of AVP/POMC/AgRP neurons in Figures 4/S4/S5. A Suppl. table (or source data) should be provided for genes in Fig. 5 and for the interaction results in Fig. 6

Reviewer #2 (Remarks to the Author):

The manuscript by Huang and colleagues examines the impact of dietary changes during lactation on offspring hypothalamic transcriptional signatures. This is an important step forward toward understanding the "transmission" of metabolic health from mother to offspring in a rather critical life period. This is a valuable resource for the field. Notably, the manuscript highlights sex-specific alterations, which have been relatively underexplored. The use of single nucleus transcriptomics allows for a deeper understanding of cellular responses, paving the way for potential interventions to mitigate the risk of obesity in offspring exposed to unhealthy maternal diets.

I would be fantastic to have had a parallel group of animals running from which comprehensive metabolic and behavioral phenotyping was done (throughout youth, adulthood, and aging) to gauge the possible links between the hypothalamic transcriptome and metabolic and behavioral outcomes, but this is clearly not a subtle undertaking.

For figure 1G, 1J, and Figure S1 please display single values in addition to the bar graph.

There there are a few typographical errors in the abstract (e.g., "matenal" should be "maternal," "celluar" should be "cellular").

Reviewer #3 (Remarks to the Author):

This study by Huang et al., investigated the impact of maternal diet on the offspring's hypothalamus. Mice were subjected to either a low or high-fat diet (HFD) during lactation, and the hypothalamus of the offspring was examined at peak lactation using single nucleus transcriptomic profiling. Key findings include:

1. There was an expansion in neuronal subpopulations in male offspring when mothers were on HFD. These include histaminergic (Hdc), arginine vasopressin/retinoic acid receptor-related orphan receptor beta (Avp/Rorb), and agouti-related peptide/neuropeptide Y (AgRP/Npy) neurons.
2. Increased interactions between Npy neurons and astrocytes were observed in offspring due to maternal overnutrition.

The authors concluded that their results highlight that maternal diet during lactation induces sex-specific cellular alterations in the offspring's hypothalamus, potentially predisposing them to metabolic disorders later in life.

Research on rodents over the past decades has revealed that elevated maternal dietary fat consumption during pregnancy and/or lactation can increase the risk of obesity for offspring in their later life, along with other significant health implications for the subsequent generation. This manuscript offers additional insights and understanding into this pertinent area of study.

- Investigating sex differences in animal models of human diseases is crucial. While the authors emphasize in the introduction (lines 72-75) that most studies predominantly focus on male offspring, it's important to note that numerous other publications have examined both sexes and the authors cite only a few papers. This is particularly true in the context of understanding the heightened risk of metabolic syndrome. For example, work done by the groups of Michael Gross, Peleg-Raibstein, Margaret J Morris, Tracy L Bale, Emily Alejandr, and many more have looked at sex differences and were not mentioned.

I have a number of methodological reservations, which I have detailed below. Additionally, there are omissions in the figure legends and the methods section.

- Changing the diet from LFD to HFD midway through pregnancy could stress the mother. In addition, mothers might exhibit variability in postnatal care, such as grooming, nursing, or even stress levels. How was this aspect managed or factored into the study?
- What percentage of each litter consisted of male and female offspring?
- How long was the male present in the cage during mating?
- Controlling offspring sampling from independent litters is crucial in experimental designs. Littermates share a common intrauterine environment and often postnatal environment as well. This shared environment can introduce a "litter effect," where the variation between litters is greater than the variation within a litter. By sampling from multiple litters, this effect can be mitigated and will ensure that the observed differences are likely due to experimental conditions and not intra-litter similarities. How many separate litters were used for all evaluations, and how many offspring were taken from each litter? This information is vital for interpreting the findings.
- I couldn't locate details regarding the number of mothers subjected to LFD versus HFD.
- Additionally, how many male and female offspring were used for metabolic measurements and the single nucleus RNA-seq experiments?
- sample size should also be mentioned in the figure legends.

We are grateful to the comments from the reviewers and we have revised the manuscript as suggested by the reviewers and included a point-to-point response to the reviewers' comments.

REVIEWER COMMENTS

Reviewer #1 (Remarks to the Author):

The manuscript addresses the interesting topic of maternal programming in the development of metabolic diseases. Previous studies have highlighted the role of maternal high fat diet feeding in mice, specifically during lactation to promote obesity and impaired glucose metabolism in the offspring. Interestingly, male offspring is particularly vulnerable to the adverse effect of maternal HFD feeding in these models. Nevertheless, the underlying neuronal mechanisms underlying these effects remain only partially defined. The authors have employed sn sequencing of hypothalamic tissue in the offspring of mice exposed to maternal HFD feeding during lactation. They report gender specific effects of the intervention with respect to cell population abundance as well as AgRP-neuron/astrocyte interactions. While the topic of the study is interesting and the sn sequencing data appear to be of adequate quality and potentially great value, the limited exploration of this unique resource does not yield the expected mechanistic insights.

Specifically, the major concerns relate to:

1. The phenotypic characterization of animals undergoing the dietary intervention (Fig. 1) is sound and largely represents what has been previously published. However, Figure labelling and the text do not match, which makes it hard to follow the result section.

RE: We appreciate the reviewer's comments and apologize for any confusion this may have caused. We have reviewed the order of the figures through the manuscript to ensure that it corresponds precisely.

2. The data represented in Fig 3D do not add much relevant information to the actual question of the work.

Re: The functionality of the distinct neuron subpopulations can be better understood with the aid of Fig. 3d, which illustrates the key metabolic pathways associated with the relevant neuronal subpopulation, including the less extensively studied Avp/Rorb neurons in relation to food intake and insulin related pathways compared to AgRP and POMC neurons. Other neuronal subpopulations that not listing in Fig.3d are not associated with such pathways. This figure will make it easier for the readers to connect the functional roles of the different types of neurons that have been described in the following figures. We have reworded the sentences (line 262 to 263) in the manuscript to make it clearer.

3. The approach to identify differential abundance with DA-seq could be a good choice to look into neuron subclusters. However, given the known difficulties with inter-sample variation when analyzing differential abundances of cell types in sc-seq data, the authors should clarify whether replicate information was included in the DA-seq analysis. If not, possibly include it and/or report results with another well-established approach such as scCODA. Furthermore, it would generally be helpful to report the cell type composition (specifically for those cell types that show differences) in cell counts/percent on a replicate level similar to Figure S2C (but for all relevant comparisons and cell type levels). In addition, in Figure S2C it is not clear which replicate comes from which experimental group. Overall, this would help to make the abundance analyses more convincing.

RE: We thank the reviewer's suggestion. The information on replicates was incorporated into the DA-seq analyses by including the replicate information in labels and the corresponding code has been uploaded to GitHub (https://github.com/hyhy200g/MouseHypothalamus_snRNA-seq). We have now reworded the sentences (line 618) in the Methods 'Differentiation abundance testing' part to clarify.

We have also applied scCODA (v. 0.1.9) in a pair-wise comparison manner, as suggested by the reviewer (**Response Fig1**). Our analyses confirmed that the mHFD group exhibited an enrichment of the AgRP, Avp/Rorb, and Hdc neurons, as demonstrated by DA-seq in the manuscript. As Lhx1 was only enriched in a small fraction of the subcluster according to DA-seq and was insignificant by scCODA, the Lhx1 subcluster enrichment has been removed from the table under Fig. S2e. The enrichment of clusters in other comparisons was also highlighted by red arrows, which exhibited consistent signals by both DA-seq and scCODA. Fig. S2C (now referred to as Fig. S2c) now includes group labelling and Fig. S3c is now at the replicate level.

Response Fig. 1 Compositional analyses using scCODA

4. Cell type abundance differences should be confirmed by independent approaches, such as in situ hybridizations and quantitative analyses of cell type representation comparing controls and dietary intervention groups.

RE: We thank the reviewers for this suggestion. We have extensively studied and validated the fraction the Npy/AgRP and POMC neurons' change and impact on neurogenesis in offspring hypothalamus combined with metabolic phenotyping and immunohistochemistry in an independent study (Xu et al., 2023). We have shown that male offspring show a larger increased ratio of NPY to POMC neurons in early postnatal neurogenesis, in response to maternal HFD during lactation.

Given the amount of work we have done in validating AgRP/Npy neurons, we will study the expansion of Avp/Rorb and Hdc neurons in male offspring in response to the maternal HFD separately in our future studies to comprehensively understand the molecular mechanisms of how these neurons been impacted by the maternal diets during lactation. We have also toned down our conclusion on the enrichment in the discussion part to highlight the further work are required to further validate the Avp/Rorb and Hdc neurons (line 485-486).

5. Regarding differential gene expression analyses, the authors implemented a workflow as recommended by Seurat for marker genes of clusters and maybe also for differentially expressed genes per group although this remains unclear from the method description. It would be nice to also include the biological replicate in the MAST model if possible (or state in the methods that it already was included). Furthermore, the authors do not really report the results of a DEG analysis (although it being the very core of what the data can provide besides the DA). Only a handful of genes are shown with violin plots (Fig. 4D). But important questions would include, which cell types are perturbed (i.e., in a simple approach: show strong are the changes (in terms of number, type of DEGs)), and is it possible to look into enriched GO terms for these cell types. In the Figures one could visualize more than 2-3 genes by using heatmaps/volcanoplots of the DEGs of the most relevant cell types (Agrp, POMC, AVP/Rorb, whatever else can be identified through DEG and/or DA analyses). Additionally, the authors should then also more systematically evaluate DEG differences between males and females.

RE: We implemented the Seurat 'MAST' method with biological replicates to calculate differentially expressed genes (DEGs), and included these details in the method section (line 622-628). We have included the dotplots of differentially expressed genes in Figure S2e and S4i and GO pathway enrichment analysis in supplemental table 4, as suggested by the reviewers. We have also included all the DEGs in AgRP, POMC, Avp/Rorb along with other key cell populations in the Supplemental table 4. We have also made corresponding changes in the manuscript and added some details of these results (line 222-238, 289-291, 333-334).

6. Regarding the subclustering provided in Fig. 4, the validity of the subclustering based on the shown marker genes remains unclear. Taking for example POMC, why do the authors not cluster by previously described marker genes such as Ttr, Anxa2, Lepr, Glp1r (Campbell et al.). It would be nice to compare these subclusterings with public data, e.g. via HypoMap to

demonstrate that they follow previously identified groupings and then connect that more clearly with the DA and DEG analyses. How does Fig. 4C relate to the previous DA-seq between mHFD and HFD findings? One could highlight those on the AGRP UMAP as well. Are there differences in DEG numbers between subtypes?

RE: We thank the reviewer for this question. We have systematically compared the POMC subclusters with previous publications (**Response Fig. 2-1**). We visualized the expression of *Pomc*, *Ttr*, *Anxa2*, *Lepr*, *Glp1r* and *Glipr1* reported from *Cambell et al* and HypoMap project in our dataset. We found the POMC clusters in our dataset express *Pomc*, but few *Ttr*, *Anxa2*, *Lepr* and *Glipr1*. One possible reason is that the number of cells in the POMC cluster in our dataset is relatively small compared to the HypoMap project. To avoid the possibility of cell capture issue, we further examined the expression of these genes in an independent dataset (Hajdarovic et al, 2022) and also found no expression of *Ttr*, *Anxa2*, and few of *Glipr1* and *Lepr*(**Response Fig. 2-1**).

Response Fig. 2-1 Expression of certain genes in POMC neurons

We further employed the hierarchical clustering to determine the correlation of POMC subclusters between our data and HypoMap, we first calculated gene score using co-expressed genes for each cell. Then we used Euclidean distance between each subcluster based on the gene score of each cell to obtain the hierarchical tree of the subclusters. We found that the our POMC cluster is more similar to Glipr1⁺ POMC cluster (C286-76) in the HypoMap dataset.

Response Fig. 2-2 hierarchical clustering of POMC subclusters

We also applied the hierarchical clustering to the AgRP cluster and found the AgRP cluster in our data has the highest correlation with the Lepr⁺AgRP⁺ cluster (C286-178) in the HypoMap dataset.

Response Fig. 2 -3 hierarchical clustering of AgRP subclusters

7. While it is nice to show the differentiation of Oligos and the differences between GFap⁺ and GFap⁻ Astrocytes (Fig. 5), this Figure is not really relevant to the maternal diet question and the different conditions are not considered in this analysis at all, aside from the statement that there were no differences.

RE: Thank you for your comment. Despite no significant differences were observed in this dataset of oligodendrocytes and astrocyte in response of maternal HFD, these differentiation trajectories recapitulate the developmental trend in non-neuronal cell types (as we showed in Fig. 5d). This further shows our dataset serves as a good resource for the community.

8. The authors claim that AGRP and astrocytes have strong interactions, but it is not clear from any plot shown in Fig. 6/S6 how they conclude that. Generally, it would be nice to not only

show which signaling molecules drive the interactions but also which cell types interact with each other (and which interactions change). In addition, while the authors employ NPY reporter mice to study potential AgRP-neuron/astrocyte interactions, these sections could also be used to quantify NPY neurons across experimental groups to provide evidence for the postulated changes in AgRP neuron numbers (see above)

RE: We thank the reviewer for the useful suggestions and we have restructured the results (Fig. 6) and text (line 372-373, line 395-401) accordingly. We believe this part of analyses has been strengthened by these changes. By implementation the CellChat, we have calculated the number of the total interactions by different groups and found highest cell-cell interaction numbers in the mHFD group compared with the other groups. We further compared the cell-cell interactions in different group pairs (mHFD vs fHFD, mHFD vs mLFD and fHFD vs fLFD) and showed the interaction weight in AgRP/Npy-astrocyte was significantly enriched in mHFD compared with fHFD group (Supplemental Fig. 6a) and this was more distinct when we focused on the key metabolic subclusters (AgRP/Npy, POMC, Avp/Rorb, Hdc and astrocytes) we have identified in mHFD group (Fig. 6b). This is why we have focused on the AgRP/Npy-astrocyte interactions in our immunohistochemistry staining validation. We have also included the key pathways and ligand-receptor interactions enriched in the mHFD vs fHFD in Fig. 6c-d as well as other group comparisons in Supplemental Fig. 6. The NEGR pathway and ligand-receptor pairs were enriched in mHFD group. NEGR was reported to be one of the top associated with body mass index (BMI) in previous GWAS.

In our recently published study (Xu et al, 2023), we observed that in males but not females, there is an increased ratio of Neuropeptide Y (NPY) to Pro-opiomelanocortin (POMC) neurons in the ARC in early postnatal neurogenesis, in response to maternal HFD. Consistent with our observations, the increase in the ratio of the key orexigenic/anorexigenic neurons in ARC region indicates that maternal overnutrition sets an obese tone for male offspring in the early stages of life. It is worth mentioning that despite the majority of AgRP and Npy neurons in the mouse hypothalamus colocalizing in the ARC region and most of the NPY^{ARC} neurons coexpress AgRP, Npy neurons have also been found in some other hypothalamic regions. For example, Npy (that colocalize with Npw in DMH) exist independent with the AgRP^{ARC} (Motoike et al., 2015). Additionally, recent study revealed that in the ARC region, a subset of AgRP negative NPY neurons are specifically activated under energy surplus conditions, which drives food intake and enhance obesity even at lower NPY levels (Qi et al., 2023). Therefore, to only use the number of Npy in whole hypothalamus or ARC region to represent the AgRP neuronal expansion might not be entirely accurate.

9. Finally, the manuscript lacks supporting data with in relation to numerous aspects of the study. Table S3 should also show percentages of cell types etc. per replicate and experimental group. While Table S4 reports the marker genes per group, the actual DEGs (between experimental groups per cluster) should be reported too. There should be a Suppl. table (or source data) for marker genes of the subclustering of AVP/POMC/AgRP neurons in Figures 4/S4/S5. A Suppl. table (or source data) should be provided for genes in Fig. 5 and for the interaction results in Fig. 6

RE: Thank you for this comment. We have now included all the relevant information into Supplemental tables. Details have been listed below:

Supplemental Table 4: Marker genes of major clusters and neuronal subpopulations as well as DEGs/GO between different maternal dietary groups.

Supplemental Table 5: Genes that related to pseudotime trajectory in astrocytes.

Supplemental Table 6: Significant ligand-receptor pairs in four maternal dietary groups.

Reviewer #2 (Remarks to the Author):

The manuscript by Huang and colleagues examines the impact of dietary changes during lactation on offspring hypothalamic transcriptional signatures. This is an important step forward toward understanding the “transmission” of metabolic health from mother to offspring in a rather critical life period. This is a valuable resource for the field. Notably, the manuscript highlights sex-specific alterations, which have been relatively underexplored. The use of single nucleus transcriptomics allows for a deeper understanding of cellular responses, paving the way for potential interventions to mitigate the risk of obesity in offspring exposed to unhealthy maternal diets.

We sincerely thank reviewer #2 for their positive comments.

1. I would be fantastic to have had a parallel group of animals running from which comprehensive metabolic and behavioral phenotyping was done (throughout youth, adulthood, and aging) to gauge the possible links between the hypothalamic transcriptome and metabolic and behavioral outcomes, but this is clearly not a subtle undertaking.

RE: Thank you for the great suggestions. For years, we have been mapping the metabolic and physiological phenotyping combine with multiomic profiling (transcriptome in offspring hypothalamus, miRNA in mother milk, lipidomics in offspring hypothalamus and mother milk with controlled diets) to understand the mechanisms of offspring in response the maternal overnutrition during lactation comprehensively. We have published a series of manuscripts in the relevant topics. In our three recent studies (Huang et al., 2020; Huang et al., 2021, Xu et al, 2023) we deeply profiled the metabolic and physiological phenotypes (including behavior observation studies during lactation of their mothers) of male and female offspring mice along with their hypothalamic transcriptomic alternations in response to different levels of maternal dietary fat (with controlled fat compositions), from lactation period all the way down to their adult phases (30 wks of age), and revealed the sex dependent mechanism in response to the maternal high fat exposure. In Xu et al., 2023, we combined bulk RNA-seq, miRNAs and lipidomics in the offspring hypothalamus and/or mother milk coupled with imaging approach and identified key sex specific genes, pathways, lipid modules and mechanisms in the offspring to response to maternal overnutrition. In the current study, by leveraging the single-nucleus RNA-seq, we characterized the sex-specific transcriptomic landscape in offspring hypothalamus and further identified mechanisms of the offspring in response to the maternal dietary effects at the single cell resolution. Regarding mapping the possible links between the

hypothalamic transcriptome and metabolic and behavioral outcomes in aging mice that have exposed to maternal overnutrition in early life, we appreciate your suggestion, and agree that this would be a very interesting angle to evaluate, however it would need a tremendous amount of effort.

2. For figure 1G, 1J, and Figure S1 please display single values in addition to the bar graph.

RE: Thank you for this comment. We have added the relevant stats in Supplemental Table S1 accordingly.

3. There there are a few typographical errors in the abstract (e.g., "matenal" should be "maternal," "celluar" should be "cellular").

RE: We apologize for the typos. We have thoroughly reviewed the abstract and manuscript and made any necessary changes.

Reviewer #3 (Remarks to the Author):

This study by Huang et al., investigated the impact of maternal diet on the offspring's hypothalamus. Mice were subjected to either a low or high-fat diet (HFD) during lactation, and the hypothalamus of the offspring was examined at peak lactation using single nucleus transcriptomic profiling. Key findings include:

1. There was an expansion in neuronal subpopulations in male offspring when mothers were on HFD. These include histaminergic (Hdc), arginine vasopressin/retinoic acid receptor-related orphan receptor beta (Avp/Rorb), and agouti-related peptide/neuropeptide Y (AgRP/Npy) neurons.
2. Increased interactions between Npy neurons and astrocytes were observed in offspring due to maternal overnutrition.

The authors concluded that their results highlight that maternal diet during lactation induces sex-specific cellular alterations in the offspring's hypothalamus, potentially predisposing them to metabolic disorders later in life.

Research on rodents over the past decades has revealed that elevated maternal dietary fat consumption during pregnancy and/or lactation can increase the risk of obesity for offspring in their later life, along with other significant health implications for the subsequent generation. This manuscript offers additional insights and understanding into this pertinent area of study.

We sincerely thank reviewer #3 for the positive comments.

- Investigating sex differences in animal models of human diseases is crucial. While the authors emphasize in the introduction (lines 72-75) that most studies predominantly focus on male offspring, it's important to note that numerous other publications have examined both sexes and the authors cite only a few papers. This is particularly true in the context of understanding the heightened risk of metabolic syndrome. For example, work done by the groups of Michael Gross, Peleg-Raibstein, Margaret J Morris, Tracy L Bale, Emilyn Alejandr, and many more have looked at sex differences and were not mentioned.

RE: Thank you for your comment. We have reworded the relevant sentences and referenced the relevant papers in response to the reviewers' comments.

I have a number of methodological reservations, which I have detailed below. Additionally, there are omissions in the figure legends and the methods section.

- Changing the diet from LFD to HFD midway through pregnancy could stress the mother. In addition, mothers might exhibit variability in postnatal care, such as grooming, nursing, or even stress levels. How was this aspect managed or factored into the study?

RE: Thank you for the comment. We didn't switch halfway through pregnancy, but change LFD to HFD at lactating day 1 (the day after delivery will be counted as lactating day 0). Our previous study showed that switching diet at lactation day 1 did not affect various aspects of maternal care in Huang et al, 2020.

- What percentage of each litter consisted of male and female offspring?

RE: For maternal LFD groups, there are 52.8% male pups and 47.2% female pups, for HFD groups, these are 47.2% male pups and 52.8% female pups, the average litter size for males (n = 3.1) and female (n = 2.8) in maternal LFD groups, male (n = 2.8) and female (n = 3.2) in maternal HFD groups. Overall, we observed well balanced male and female offspring number in all the groups.

- How long was the male present in the cage during mating?

RE: Initially we housed the males with females for one week and then removed the males from the cage, if the females did not have the sign of pregnancy in the following week (we closely monitored the female body weight, the gradually increase of the BW was considered as a sign of pregnancy), we put the males back to the cage with females for another week. If the females were not pregnant after the 2nd mating, we removed that individual from our experiment.

- Controlling offspring sampling from independent litters is crucial in experimental designs. Littermates share a common intrauterine environment and often postnatal environment as well. This shared environment can introduce a "litter effect," where the variation between

litters is greater than the variation within a litter. By sampling from multiple litters, this effect can be mitigated and will ensure that the observed differences are likely due to experimental conditions and not intra-litter similarities. How many separate litters were used for all evaluations, and how many offspring were taken from each litter? This information is vital for interpreting the findings.

RE: Thank you for your comment. For the metabolic measurements, 9 independent litters from the maternal LFD group and 6 from the maternal HFD group were used. We measured all the offspring in each litter (5 or 6 offspring for each litter). In the single-nucleus RNA-seq experiments, we pooled 2 offspring of the same sex from each litter and two independent litters were used for each experimental group. We also corrected the potential batch effect by adding "sampleID" as a covariate during data integration and confirmed that there was no sample or batch effect by visualization the cell distribution on the UMAP (shown in Figure S2b). After batch effect correction, the "litter effect" has also been corrected and would not affect the downstream analysis.

- I couldn't locate details regarding the number of mothers subjected to LFD versus HFD.

RE: Sample sizes of mothers fed with LFD were 9, for mothers with HFD were 6. We have added those numbers accordingly in the figure legend for Fig. 1 as well as in the main text.

- Additionally, how many male and female offspring were used for metabolic measurements and the single nucleus RNA-seq experiments?

RE: All male and female offspring from all the litters were used for metabolic measurements across the lactation. For the experimental group of single nucleus RNA-seq experiment (four groups in total: fLFD, fHFD, mLFD, mHFD), we pooled two male or two female offspring from each litter for every sample, we collected duplicates for each of the group. A total of 8 samples from 8 different litters were collected.

- sample size should also be mentioned in the figure legends.

RE: We have now revised the figure legends thoroughly and added the sample size as suggested by the reviewers.

References

Hajdarovic, K.H., Yu, D., Hassell, L. et al (2022). Single-cell analysis of the aging female mouse hypothalamus. *Nature Aging*. 2, 662-678

Huang, Y., Osorio Mendoza, J., Li, M., Jin, Z., Li, B., Wu, Y., Togo, J., and Speakman, J.R. (2021). Impact of graded maternal dietary fat content on offspring susceptibility to high-fat diet in mice. *Obesity (Silver Spring)*. 29, 2055-2067.

Huang, Y., Mendoza, J.O., Hambly, C., Li, B., Jin, Z., Li, L., Madizi, M., Hu, S., and Speakman, J.R. (2020). Limits to sustained energy intake. XXXI. Effect of graded levels of dietary fat on lactation performance in Swiss mice. *J Exp Biol*. 223.

Motoike, T., Skach, A.G., Godwin, J.K et al (2015). Transient expression of neuropeptide W in postnatal mouse hypothalamus – A putative regulator of energy homeostasis. *Neuroscience*. 301, 323-337.

Qi, Y., Lee, N.J., Kin Ip, C. et al (2023) *Agrp*-negative arcuate NPY neurons drive feeding under positive energy balance via altering leptin responsiveness in POMC neurons. *Cell Metab*, 35, 979–995.

Xu, Y., Yang, D., Wang, L., Krol, E., Mazidi, M., Li, L., Huang, Y., Niu, C., Liu, X., Lam, SM, Shui, G., Douglas, A., Speakman, J.R. (2023). Maternal High Fat Diet in Lactation Impacts Hypothalamic Neurogenesis and Neurotrophic Development, Leading to Later Life Susceptibility to Obesity in Male but Not Female Mice. *Adv Sci*, 2305472.

REVIEWERS' COMMENTS

Reviewer #1 (Remarks to the Author):

The authors have appropriately addressed the previous concerns-thank you!

Reviewer #2 (Remarks to the Author):

The authors have addressed all my concerns.

Reviewer #3 (Remarks to the Author):

The authors have addressed the majority of my concerns from the previous submission.

Regarding my comment:

Examining sex differences in animal models of human diseases is of utmost importance. While the authors highlight in the introduction (lines 72-75) that the majority of studies primarily concentrate on male offspring, it is crucial to acknowledge that a considerable body of literature has investigated both sexes, and the authors have cited only a limited number of these studies...

The authors responded by stating that they had rephrased the relevant sentences and included references to the pertinent papers in response to the reviewers' feedback. However, I could not discern these changes in the revised manuscript.

We are grateful to the comments from the reviewers, we have revised the manuscript as suggested, and included here a point-to-point response to the reviewers' comments.

REVIEWER COMMENTS

Reviewer #1 (Remarks to the Author):

The authors have appropriately addressed the previous concerns-thank you!

Reviewer #2 (Remarks to the Author):

The authors have addressed all my concerns.

RE: We sincerely thank reviewer #1 and #2 for their comments and suggestions for improving our manuscript!

Reviewer #3 (Remarks to the Author):

The authors have addressed the majority of my concerns from the previous submission.

Regarding my comment:

Examining sex differences in animal models of human diseases is of utmost importance. While the authors highlight in the introduction (lines 72-75) that the majority of studies primarily concentrate on male offspring, it is crucial to acknowledge that a considerable body of literature has investigated both sexes, and the authors have cited only a limited number of these studies...

The authors responded by stating that they had rephrased the relevant sentences and included references to the pertinent papers in response to the reviewers' feedback. However, I could not discern these changes in the revised manuscript.

RE: We greatly appreciate the reviewer's comments and apologize for any confusion this may have caused. We have now rephrased these sentences in the introduction and cited these literatures accordingly in the revised manuscript (line72-73).